# Unlocking SLM Potential for Data Analysis Code Generation via Non-Parametric Knowledge Distillation

**Jinyang Li**$^{\alpha\beta}$,*, **Jack Williams**$^{\alpha}$, **Nick McKenna**$^{\alpha}$, **Arian Askari**$^{\alpha}$,
**Nicholas Wilson**$^{\alpha}$, **Reynold Cheng**$^{\beta}$

$^{\alpha}$Microsoft Research Cambridge    $^{\beta}$The University of Hong Kong
jl0725@connect.hku.hk, jack.williams@microsoft.com,
ckcheng@cs.hku.hk

## Abstract

Knowledge distillation from Large Language Models (LLMs) to locally hosted Small Language Models (SLMs) provides advantages for Data Analysis Code Generation (DACG) such as privacy protection. However, achieving effective distillation without resource-intensive training is challenging. This paper investigates whether LLMs can distill knowledge to SLMs through In-Context Learning (ICL), a training-free method for rapid task adaptation. We present the **DARGO**: *Distillation and Adaptive Reasoning-Guided Orchestration* framework, which facilitates automatic knowledge distillation from LLMs to SLMs. DARGO consists of three phases: exploration through an **Model Orchestration Interface (MOI)**, **Memory Collection** of successful trajectories, and **Knoweldge-driven Inference**. We evaluate DARGO on three challenging DACG benchmarks (WIKITQ, TABMWP, and BIRD-SQL), each with in-domain training sets that enable detailed analysis of knowledge distillation effectiveness. DARGO demonstrates a substantial relative performance improvement of 27.5% on average for the student SLMs. To further observe generalization capabilities, we evaluate the DARGO across different teacher-student model combinations, knowledge transfer scenarios, and unified memory approaches for more advanced, test-only data analysis tasks. Our findings contribute a novel perspective on distillation methods that enhance performance for SLMs while avoiding intensive fine-tuning. The source code is available: https://github.com/accpatrick/DarGO.

## 1 Introduction

Data Analysis Code Generation (DACG) automates the conversion of natural language queries into executable code, empowering information extraction and analysis from tabular data efficiently. This process enhances productivity, reduces the technical barrier for data analysis, and allows data scientists to focus on deriving insights, ultimately supporting more effective decision-making [24, 17, 14]. This is a challenging task since it not only requires the capability of code generation but also understanding complex tabular data.

Large Language Models (LLMs) have demonstrated remarkable performance across diverse, complex tasks [49, 37, 8, 71, 11]. Leveraging LLMs or LLM agents for automatic code generation from user queries offers an effective solution [65, 56]. However, the integration of LLMs in DACG faces two primary challenges: 1) Privacy concerns arise when utilizing closed-source LLMs such as GPT-4 [2] or Claude-3.5-Sonnet [40]. 2) Deploying large open-source models like Llama-3.1-405B [13] or DeepSeek-v3 (671B) [31] can be challenging due to their large number of parameters. Balancing these benefits and challenges is crucial for effective data science applications.

---

*Work done during Research Intern at Microsoft Research Cambridge

39th Conference on Neural Information Processing Systems (NeurIPS 2025).

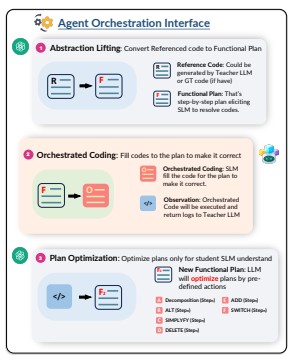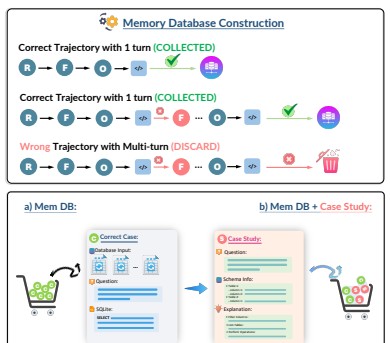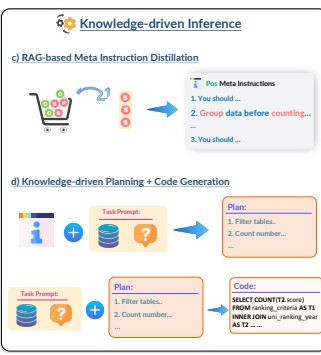

Figure 1: Overview of the orchestration system of DARGO for data science code generation. Left: Model Orchestration Interface (MOI) with abstraction lifting, orchestrated coding, and plan optimization. Center: Memory Database Construction, including trajectory collection and case study integration. Right: Knowledge-driven Inference and Planning, featuring RAG-based meta instruction generation, and knowledge-driven code generation. **R:** Referenced GT Code. **F:** Functional Plans. **O:** Orchestrated Coding.

Small Language Models (SLMs), such as Phi-3-mini [1] and Llama-3.1-8B [13], have gained attention for their In-Context Learning (ICL) capabilities and ability to be locally deployed. These models offer computational efficiency and enhanced data privacy, crucial for resource-constrained or privacy-sensitive applications [21]. While SLMs have shown competitive performance in some general tasks including natural language understanding [39] and code completion [7], their effectiveness in data science code generation tasks remains an open question.

Fine-tuning is a common strategy to enhance SLM capabilities for complex tasks [42]. However, this approach encounters several challenges in the domain of data science DACG. One primary issue is the limited availability of high-quality training data. Professional tabular datasets, such as relational databases, are often small or proprietary, restricting access to substantial corpora for training. Additionally, the dual expertise required in both coding syntax and data understanding for accurate annotation further constrains dataset scalability [30, 28]. This is reflected in recent benchmarks for data science code generation, which typically contain around or fewer than 1,000 samples, highlighting the complexity and resource constraints in this field [20, 3, 26, 70, 66]. Recent research has explored distillation from LLMs to SLMs through fine-tuning on synthetic data [52, 36, 22]. While this approach shows promise, several challenges persist. For example, the performance improvements obtained from fine-tuning approaches can fail to generalize across different programming languages or dialects, requiring re-training for each package update or new task [46, 23]. However, In-Context Learning (ICL) can adapt to new requirements or tasks by providing relevant instructions or examples, reducing the effort required for re-training or continual training. This raises our central research question in DS code generation: ***Can LLMs distill knowledge to SLMs through In-Context Learning (ICL)?***

In this paper, we explore the potential of knowledge distillation from LLMs to SLMs via ICL. We propose a novel **D**istillation and **A**daptive **R**easoning-**G**uided **O**rchestration framework that enables an LLM to serve as a Teacher model guiding SLMs (Student models) in complex DACG tasks. DARGO operates in three phases: **exploration**, **memory database collection**, and **knowledge-driven inference**. During exploration, we employ the Model Orchestration Interface (**MOI**) that allows an LLM to probe and analyze SLM code knowledge by converting questions into step-wise functional plans and asking SLMs to infill the code for each plan. Then, successful collaborated cases are stored in a memory databases. In the knowledge-driven inference phase, DARGO utilizes a retrieval-augmented generation (RAG) approach that dynamically distills knowledge at inference time by presenting relevant prior successful cases in a case-study format to guide the generation process, which proves friendly to SLMs to absorb (Section 2.3).

We evaluate DARGO on three challenging tabular analysis benchmarks requring code generation:TABMWP [34], BIRD-SQL [29], and WIKITQ [41], where each feature fixed, non-overlapping train–test splits to closer show how effective of knowledge distillation that DARGO is. The experimental results demonstrate that the DARGO framework significantly improves the performance

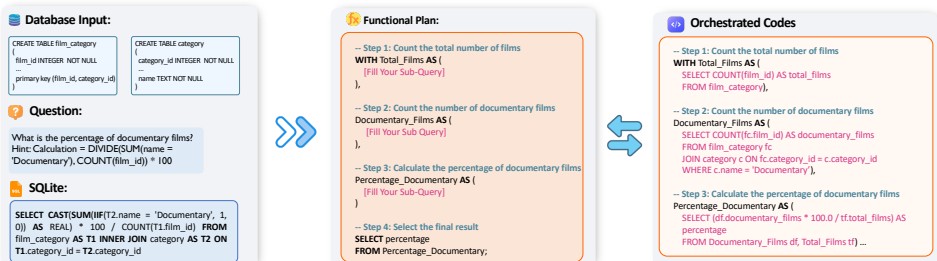

Figure 2: Main steps of MOI demonstrated with a text-to-SQL task example. Teacher model converts referenced code to a functional plan for Student models to complete. Teacher model iteratively optimizes the plan until the Student model produces correct code. A Python code MOI example is provided in Appendix D.1.

of SLMs across all datasets, validating the potential of our knowledge distillation approach via ICL. Importantly, the memory produced for one student can guide *other* students, demonstrating model-agnostic transfer. Additional cross-dataset experiments on CRT-QA [70], QRDATA [32], and Infi-Agent [20] confirm that the distilled knowledge generalizes beyond the original training distribution. Taken together, our results show that lightweight, privacy-preserving SLMs can inherit much of an LLM's analytic expertise through ICL alone, making DACG practical in resource-constrained or privacy-sensitive environments.

## 2 Methodology

### 2.1 Task Formulation

Given a natural language query $q_i \in \mathcal{Q}$, where $\mathcal{Q} = \{q_1, q_2, \ldots, q_N\}$ represents a set of $N$ queries, the corresponding tabular data or database schema information $d_i \in \mathcal{D}$, where $\mathcal{D} = \{d_1, d_2, \ldots, d_N\}$, then the Small Language Model (SLM) is tasked with generating an executable code snippet $c_i$. This code snippet must accurately answer the query $q_i$ with the associated data $d_i$. The function that maps each query-data pair to its corresponding code snippet by SLMs is defined as $f_{\text{gen}}$, and can be written as:

$$c_i = f_{\text{gen}}(d_i, q_i) \quad \text{for } i = 1, 2, \ldots, N. \tag{1}$$

### 2.2 Model Orchestration Interface

The Model Orchestration Interface (MOI) is conducted between a Large Language Model (LLM), a Teacher model, and a Student model, represented by the SLM. A Teacher LLM with superior reasoning capabilities will often generate plans that are too abstract for a weaker Student SLM to enact. These plans require decomposition and refinement to match the operational granularity of the Student SLM. Through orchestrated mediation, the MOI dynamically adjusts plan granularity to align with the SLM's code capabilities whilst maintaining task conformance. The MOI is composed of three key components: Abstraction Lifting, Orchestrated Coding, and Plan Optimization.

**Abstraction Lifting (AL).** In this phase, LLM generates a functional plan $P_i = \{s_{i1}, s_{i2}, \ldots, s_{iK}\}$ based on a query $q_i$, data input $d_i$, and the corresponding ground truth (gt) code $\tilde{c}_i$. The ground truth code $\tilde{c}_i$ can either be sourced from an existing dataset (BIRD-SQL) or generated by the Teacher model when it is not directly available but a ground-truth answer string exists, such as WIKITQ and TABMWP. This functional plan is defined as $\mathcal{L}_{\text{al}}(d_i, q_i, \tilde{c}_i)$, where $\mathcal{L}_{\text{al}}$ denotes the LLM performing abstraction lifting. Each step $s_{ij}$ in the plan corresponds to a key subtask derived from the query, collectively forming a structured template outlining the solution process. These steps are annotated by the LLM with descriptive comments and placeholders such as [Fill Your Code Here] in Python or [Fill Your Sub-Query] in SQL, as shown in Figure 2, ensuring that the SLM follows the logical flow of the entire plan and enables guided code generation. Unlike Chain-Of-Thought [60] plans, which provide intermediate steps in continuous textual form, our approach

Table 1: The 6 action types utilized by the LLM during the Model Orchestration Interface (MOI) to optimize the plans for better understanding and code generation by SLMs

| Action Type | Expression | Description |
|---|---|---|
| **Decomposition** | `step(x) → step(a),`
`step(b)` | Split a complex step `x` into smaller, manageable steps such as step `a` and step `b`. |
| **ALT** | `step(x) → step(y)` | Replace a step `x` described by ambiguous or incorrect messages with a clearer and correct alternative step `y`. |
| **ADD** | `step(x) → step(x),`
`step(a)` | Add a necessary step `a` to ensure the completeness of code logic. |
| **DELETE** | `step(x) → None` | Remove the unnecessary step `x`, which may lead to misunderstanding by the SLM. |
| **SIMPLIFY** | `step(x) →`
`simple_step(x)` | Replace a complex step `x` with a simpler approach. For example, convert recursive plans into iterative loops. |
| **SWITCH** | `packageA.step(x) →`
`packageB.step(x)` | Use a simpler package to achieve the same functionality. For example, conversion from Package `Linear Regression` to `Correlation Coefficient` to determine relationship between two variables. |

bridges high-level problem understanding with low-level code implementation logic, allowing the SLM to more effectively follow the plan for DACG.

**Orchestrated Coding (OC).** Once the functional plan $P_i$ is provided, the Student SLM considers all context including the question and data input to generate the complete orchestrated code $c_i = f_{\text{gen}}(d_i, q_i, P_i)$ in a single turn by filling all placeholders, ensuring the solution is correct and executable. The results from executing this orchestrated code are then compared to those from a reference solution (such as ground truth answer string or gt codes) to evaluate whether the SLM fully understands both the data and the logic needed to answer the question. This comparison serves as a key indicator of the problem-solving accuracy of SLM and alignment with the intended solution. While the ground truth code may already be available from datasets or generated by the Teacher model, orchestrated coding and abstraction lifting are crucial for a few reasons. **First**, AL breaks down complex problem-solving tasks into manageable sub-tasks, with the potential to improve the performance of the SLMs across a wide range of analytical queries by assisting them in understanding modular structure. **Additionally**, error isolation can be grounded in the program structure, enabling more precise identification of issues and contributing to optimized plans. This is supported by our analysis in Section 3.6 that compares chain-of-thought with functional plans for exploration.

**Plan Optimization (PO).** The plan optimization process is an iterative procedure that unfolds over multiple turns, denoted by $t$. During each iteration, the SLM refines the functional plan $P_i^t$. To formalize this interactive optimization process, we define an environment $\mathcal{E} = \langle \mathcal{S}, \mathcal{A}, \mathcal{O}, \mathcal{T} \rangle$, following [72, 62, 16], where $\mathcal{S}$ represents the state space, $\mathcal{A}$ the action space (Table 1), and $\mathcal{O}$ the observation space. In this context, the plan $P_i^t$ is embedded within the current state $\mathcal{S}_i^t$, serving as a structure that guides the SLM to generate code. The orchestrated code $c_i^t$ is the snippet produced by performing the plan $P_i^t$ within the environment.

During each turn, the LLM observes $o_i^t$, the outcome of executing orchestrated code $c_i^t$ generated by the SLM, and selects an action $a_i^t$ from $\mathcal{A}$ to optimize the plan. For example, if a step $s_{ij}^t$ contains an error such as: `"Step j:  List players who was born before 1930 and after 1950"`, the LLM will apply an action `ALT(·)` to correct this, resulting in an updated plan $P_i^{t+1}$ with the refined step $s_{ij}^{t+1}$ as `"Step j:  List players who were born after 1930 and before 1950"`. This iterative process can be represented as:

$$P_i^{t+1} = \{s_{i1}^t, s_{i2}^t, \ldots, s_{ij}^{t+1}, \ldots, s_{iK}^t\},$$
$$s_{ij}^{t+1} = \mathcal{L}_{\text{opt}}\left(s_{ij}^t, o_i^t, a_i^t\right). \tag{2}$$

Here, $s_{ij}^{t+1}$ is updated by the optimization function $\mathcal{L}_{\text{opt}}$ of the LLM, which integrates the current observation $o_i^t$, action $a_i^t$ and sub-optimal step $s_{ij}^t$. The system transitions from state $\mathcal{S}_i^t$ to $\mathcal{S}_i^{t+1}$ through $\mathcal{T}(\mathcal{S}_i^t, \mathcal{A}_i^t)$, resulting in the updated plan $P_i^{t+1}$. The SLM then generates the updated orchestrated code $c_i^{t+1} = f_{\text{gen}}(q_i, d_i, P_i^{t+1})$ for the new plan. This process repeats until the output is correct or the maximum number of iterations $T$ is reached.

## 2.3 Memory Database Construction

After interactions between the LLM and SLM in MOI, the finalized states are stored in a memory database. This database includes the correct orchestrated codes, along with the context of the question and data input. This process ensures that the SLM can efficiently reference and apply related knowledge to new, unseen queries.

```
### Case Study: Average Weight Calculation for Specific Players

### Question:
What is the average weight of Jamarr Sanders and Robert Williams?

### Table Info:
- **Columns**: Name, Height, Weight (lbs.), Position, Class, Hometown, Previous Team(s)
- **Sample Data**:
  - Jamarr Sanders: Weight 210 lbs.
  - Robert Williams: Weight 210 lbs.

### Objective:
To calculate the average weight of the players Jamarr Sanders and Robert Williams from the given dataset.

### Explanation:
1. **Load Data**: The data is loaded from a tab-separated values (TSV) file.
2. **Filter Data**: Rows corresponding to the names "Jamarr Sanders" and "Robert Williams" are filtered from the dataset.
3. **Calculate Average**: The average weight of the filtered rows is computed.
4. **Output**: The result is printed as an integer.

By following these steps, the student can understand how to filter specific rows in a dataset and perform calculations on the filtered data. This case demonstrates the practical application of data manipulation and analysis using pandas in Python.
```

Figure 3: An example of a generated case study to enhance comprehension for SLMs.

**Case Study Translation.** Rather than only storing raw, heterogeneous cases that consist of a query, plan, and orchestrated code in a simple stacked format, the LLM refines these into case study-like representations. These representations distill the reasoning behind the success of each example, serving as an intermediate abstraction that emphasizes the underlying rationale for the chosen approach. Each case study $G_i$ contains a `Case Name`, `Question`, `Schema / Value Information`, `Objective`, and an `Explanation` of how the solution code successfully addresses the query using the provided data. An example of this structure is provided in Figure 3. As shown in Figure 1 (a)-(b), DARGO performs case study translation only for correct cases, because reflecting on incorrect cases without supervision often introduces hallucinations.

**Correct Case Collection.** The Correct Case Collection, denoted as $\mathcal{M}$, consists of cases where the SLM has generated correct orchestrated codes. Each case $M_i$ in this collection contains the natural language query $q_i$, the corresponding data $d_i$, the correct orchestrated code $\hat{c}_i$, which contains descriptive comments as shown in Figure 2 (right), and the case study $G_i$ illustrating the solution. The set $\mathcal{M}$ is the union of all such individual cases:

$$\mathcal{M} = \bigcup_i M_i, \quad M_i = (q_i, d_i, \hat{c}_i, G_i). \tag{3}$$

## 2.4 Knowledge Distillation from Memory Database

This part presents a RAG-based method of knowledge distillation as a global instruction, termed as meta-instruction. This instruction then guides the SLM in learning how to plan and generate code more accurately for unseen queries.

**RAG-based Knowledge Distillation for Meta Instruction Generation.** We propose a Retrieval-Augmented Generation (RAG) framework for localized instruction distillation. In the retrieval phase, we identify the top relevant examples from a memory database $\mathcal{M}$ via an embedding model. Relevance is measured via a function $\mathcal{D}$, expressed as $\mathcal{R}(q_i) = \mathcal{D}(q_i, \mathcal{M}, k)$, where $k$ is number of most relevant cases.

| Model | WIKITQ | TABMWP | | | BIRD-SQL | | | |
|---|---|---|---|---|---|---|---|---|
| | Accuracy | Grad. 1-6 | Grad. 7-8 | Total | Sim. | Med. | Chal. | Total |
| CodeLlama-7B | 11.80 | 26.55 | 13.11 | 20.50 | 43.92 | 18.00 | 11.76 | 24.40 |
| CodeLlama-13B | 34.90 | 37.27 | 24.22 | 31.40 | 45.27 | 19.60 | 17.65 | 26.80 |
| StarCoder2-7B | 20.70 | 34.00 | 27.56 | 31.10 | 41.22 | 21.60 | 17.65 | 26.60 |
| StarCoder2-15B | 36.60 | 39.09 | 33.14 | 36.50 | 43.92 | 29.20 | 14.71 | 30.60 |
| Phi-3-Small-7B | 27.00 | 46.36 | 38.00 | 42.60 | 52.03 | 28.40 | 10.78 | 31.80 |
| Phi-3-Medium-14B | 44.80 | 59.45 | 46.00 | 53.40 | 51.35 | 32.80 | 13.73 | 34.40 |
| *SLM Performance* | | | | | | | | |
| **Phi-3-Mini-3.8B** | 32.50 | 44.18 | 38.89 | 41.80 | 38.51 | 21.20 | 11.76 | 24.40 |
| + Chain-Of-Thought | 27.70 | 46.36 | 35.33 | 41.40 | 34.46 | 22.00 | 12.75 | 23.80 |
| + Static Few-Shot | 23.00 | 37.27 | 34.89 | 36.20 | 47.97 | 20.80 | 7.84 | 26.20 |
| + Dynamic Few-Shot | 16.60 | 51.45 | 52.89 | 52.10 | 42.57 | 18.80 | 11.76 | 24.40 |
| + DSPy Distillation (Few Shot) | 26.70 | 42.70 | 37.60 | 40.40 | 36.00 | 19.80 | 11.00 | 22.80 |
| + ReGAL Distillation | 36.10 | 44.73 | 38.22 | 41.80 | 35.14 | 14.80 | 12.75 | 20.40 |
| + DARGO Meta Instruction (Ours) | **41.10** | **61.27** | **56.44** | **59.10†** | **51.35** | **30.40** | **16.67** | **33.80** |

Table 2: Performance comparison of various SLMs on WIKITQ, TABMWP, and BIRD-SQL, with results presented in accuracy percentages. Improvements of our DARGO methods over the End-to-End Code Gen baseline are highlighted using different intensities of olive color. **Bold** indicates best results for Phi-3-Mini, while underlines denote second-best results.[†] means a hybrid with dynamic few-shot with DarGO. Detailed instructions of when to use few shot is in Appendix A.4.

Then, as shown in Figure 1 (d), the case studies of these retrieved examples are then fed into the SLM to extrapolate plans for solutions, adhering them to the specific query. Here, SLM performs a secondary distillation, extracting shared knowledge patterns from these case studies, which have already been distilled by the LLM (Teacher model), to generate instructions, noted as **Meta-Instruction** ($\mathcal{I}_{\mathbf{m}}(q_i)$), precisely specific to the current query at hand. The process is formalized:

$$\mathcal{I}_{\mathrm{m}}(q_i) = f_{\mathrm{agg}}(q_i, \mathcal{R}(q_i)) \tag{4}$$

where $f_{\mathrm{agg}}$ is an aggregation function applied by the SLM. By doing so, SLMs can generate more relevant and contextually appropriate instructions, effectively bridging the gap between general knowledge and query-specific requirements.

**Knowledge-Driven Inference.** Harnessing the distilled instructions $\mathcal{I} = \mathcal{I}m(q_i)$ from the memory database, the SLM initially formulates a structured plan $p_{\mathrm{gen}}$, which it subsequently employs to generate code for new queries (Figure 1 (e)). For a given query $q_i$ and its associated data $d_i$, this process unfolds as follows:

$$P_{\mathrm{gen}} = f_{\mathrm{plan}}(\mathcal{I}, d_i, q_i), \quad c_i = f_{\mathrm{gen}}(d_i, q_i, P_{\mathrm{gen}}), \tag{5}$$

where $f_{\mathrm{plan}}$ denotes the planning function executed by the SLM. This plan serves as a blueprint, guiding the following code generation phase. The SLM then employs the function $f_{\mathrm{gen}}$, which takes $P_{\mathrm{gen}}$ along with the original query $q_i$ and data $d_i$ to generate the final code $c_i$.

# 3 Experiments

In this section, we first describe datasets and evaluation metrics in Section 3.1, followed by implementation details in Section 3.2. We then present a comprehensive experiments aimed at addressing three key research questions:

- **RQ1 (Section 3.3)**: How effective is DARGO for DACG?
- **RQ2 (Section 3.4)**: Does DARGO and the knowledge it distills generalize across models?
- **RQ3 (Section 3.5)**: How does DARGO compare to popular Lora fine-tuning?
- **RQ4 (Section 3.6)**: Are all components of DARGO necessary?

## 3.1 Datasets and Metrics

We evaluate our approach on the DACG datasets WIKITQ [41], TABMWP [34], and BIRD-SQL [29]. These datasets vary in data complexity and task requirements. Full dataset statistics are provided in Appendix B.1. WIKITQ features operations such as counting, comparison, and aggregation, and we

Table 3: Zero-shot performance of DARGO on different SLMs.Improvements (in parentheses) show gains over the Baseline.

| Model | WIKITQ | TABMWP | BIRD-SQL |
|---|---|---|---|
| *StarCoder-15B as Student* | | | |
| Baseline | 36.60 | 36.50 | 30.60 |
| + DARGO-MI | 45.70 (↑9.10) | 43.93 (↑7.43) | 38.40 (↑7.80) |
| *Llama-3.1-8B as Student* | | | |
| Baseline | 34.30 | 42.90 | 40.20 |
| + DARGO-MI | 39.80 (↑5.50) | 49.10 (↑6.20) | 44.20 (↑4.00) |

Table 4: Zero-shot performance of DARGO on different teacher LLMs. → means distill knowledge from teacher LLMs to student SLMs.

| Model | WIKITQ | TABMWP | BIRD-SQL |
|---|---|---|---|
| *Llama-3.3-70B → Phi-3-mini* | | | |
| Baseline | 41.80 | 32.50 | 24.40 |
| + DARGO-MI | 48.80 (↑7.00) | 42.60 (↑10.10) | 36.20 (↑11.80) |
| *Llama-3.3-70B → Llama-3.1-8B* | | | |
| Baseline | 42.90 | 34.30 | 40.20 |
| + DARGO-MI | 50.40 (↑7.50) | 42.00 (↑7.70) | 47.00 (↑6.80) |

select 1,000 test and 2,000 training examples. Accuracy is measured using the official scripts from Pasupat and Liang [41]. TABMWP involves mathematical word problems in tabular data, with 1,000 questions used for memory construction and a separate 1,000 test set. Performance is evaluated by comparing answers to ground truth. BIRD-SQL presents relational databases with both semantic parsing and analytical tasks. We adopt the 1,000-example mini-train set from Qu et al. [44] and evaluate on a 500-example mini-dev set using execution accuracy (EX).

## 3.2 Implementations

**Setup.** We conduct experiments in two settings. For TABMWP and WIKITQ, we follow Stengel-Eskin et al. [50] to first have a Teacher LLM (GPT-4o) generate silver programs during exploration. At inference, an SLM produces Python code, which is executed to obtain a final answer string for comparison with the ground truth answer. For BIRD-SQL, no silver code is required because the dataset already contains ground-truth SQL. RAG-based Meta Instructions employ $k = 3$ nearest neighbors (via CodeT5+ [59]). Further implementation details appear in Appendix B.

**Baselines Models.** We consider an SLM suitable if it can perform in-context learning and has under 15B parameters for one A100 80G GPU inference. For closed-source models, we choose GPT-35-Turbo as SLM since it has faster inference speed and its performance falls behind advanced models (e.g., GPT-4). We classify them into three categories: *Orchestration Models*, where GPT-4o is the Teacher, and Phi-3-mini-128k [1], Llama-3.1-8B [13] serve as Students for verifying the generalization of our approach. *Evaluation Models*, including Phi-3, CodeLlama, and StarCoder2 families. *Knowledge-Transmission Models*, containing GPT-35-Turbo and Llama-3.1-8B, used to test knowledge transmission in Section 3.4.

**Baseline Methods Implementation.** Baseline methods include zero-shot end-to-end, Chain-of-Thought [60], static few-shot [6], and dynamic RAG-based few-shot [15] with memory database by DARGO since python tasks do not have GT codes but answer string, each with three examples. We also compare two distillation frameworks, DSPy [25] and ReGAL [51], both using GPT-4o as the teacher.

## 3.3 Overall Results

Table 2 highlights three key aspects:

**1) Effectiveness of DARGO:** Distilling knowledge via **DARGO** propels Phi-3-mini to outperform both End-to-End Code Generation and Chain-of-Thought on all SLM datasets, achieving relative gains of 17.5% on TABMWP to 38.5% on BIRD-SQL. Moreover, Phi-3-mini with DARGO often surpasses larger models (2–3× parameters), outperforming CodeLlama-13B by 7.0% and StarCoder2-15B by 3.2% on BIRD-SQL, and rivaling Phi-3-Medium (4× parameters) across all benchmarks.

**2) Comparison with Advanced Distillation Strategies.** We implement two advanced distillation pipelines, DSPY and REGAL, on GPT-4o and Phi-3-mini for fair comparison, reporting the best performance of REGAL across different numbers of reusable helper functions (1–10) to minimize bias. As shown in Table 2, **DARGO** outperforms both approaches on all three datasets and difficulty levels.

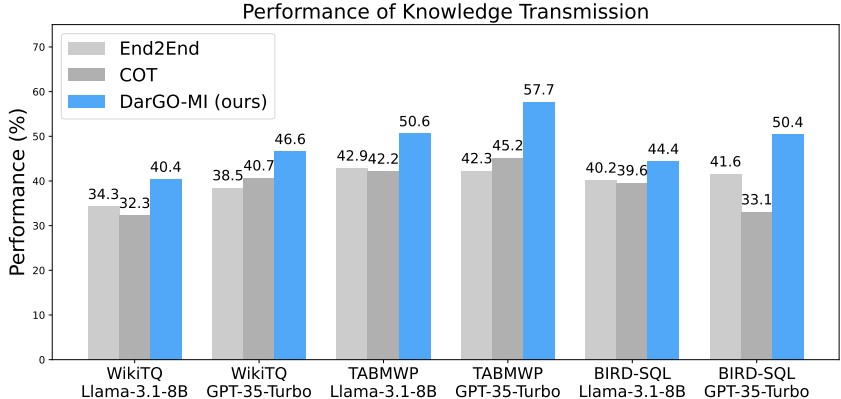

Figure 4: Knowledge transmission for from the memory database between GPT-4o and Phi-3-mini across three datasets.

DSPY is robust for general QA but struggles in execution-based DACG settings because its exact-match metric does not account for valid alternative program solutions. Moreover, DSPY relies on few-shot demonstrations rather than comprehensive knowledge distillation, reflecting the broader challenges of demonstration-based methods for complex DACG. By contrast, **DARGO** prioritizes task-specific instructions to guide complex operations, shifting focus from prompt optimization to experience summarization.

We also evaluate REGAL, which provides Reusable Python Helper Functions from the memory database. Although promising for tightly scoped, domain-specific tasks, REGAL proves less effective for cross-domain DACG (see Table 2). Error analysis shows it often produces narrowly tailored functions (e.g., `get_least_points_team`, `filter_senator_by_year`), which often directly reference the data schema and are therefore irrelevant on new tasks. In contrast, the distilled knowledge of DARGO is textual, focusing on reasoning-level challenges, and is therefore not syntactically fixed to a specific domain or input.

## 3.4 Generalization

Here, we conduct experiments to test generalization of DARGO framework:

**Is DARGO model-agnostic to SLMs?** To investigate the breadth of the DARGO pipeline beyond a single teacher–student configuration, we perform two complementary experiments. (1) We apply the complete workflow to two additional student models: `StarCoder2-15B` and `Llama-3.1-8B` while keeping `GPT-4o` as the teacher. As summarized in Table 3, both students outperform their respective baselines on all three datasets, indicating that DARGO consistently improves a variety of SLM architectures. (2) To examine robustness with respect to the teacher LLM, we replace `GPT-4o` with `Llama-3.3-70B`. The results in Table 4 replicate the earlier gains, demonstrating that gains of DARGO are not confined to a specific teacher model.

**Is the distilled knowledge only useful to the student model participating in Orchestration?** Next, we examine whether the knowledge distilled through DARGO is only beneficial to the student model involved in the orchestration phase. Even though this has been proven effective, it would be costly if exploration phase is repeated for new SLMs. Therefore, we test the distilled instructions on two new SLMs, `Llama-3.1-8B` and `GPT-35-Turbo`, both of which have not participated in the exploration between `GPT-4o` and `Phi-3-mini`.

Figure 4 shows that meta instructions (MI) derived from our memory database (jointly updated by GPT-4o and `Phi-3-mini`) produce siginificant improvements over baselines. Specifically, `Llama-3.1-8B` achieves an average relative improvement of 14.3%, while `GPT-35-Turbo` sees a 30.9% gain. These results indicate that the distilled knowledge is not tied to a specific student model; it can effectively transfer to new models without additional fine-tuning, offering a scalable means of augmenting emerging SLMs.

Table 5: Performance of DARGO on different SLMs. Improvements (in parentheses) are over the Baseline.

| Model | CRTQA | QRDATA | INFI-AGENT |
|---|---|---|---|
| *Phi-3-mini as Student* | | | |
| Baseline | 26.51 | 31.14 | 42.80 |
| + DARGO-MI | 32.12 (↑5.61) | 38.99 (↑7.85) | 46.69 (↑3.89) |
| *Llama-3.1-8B as Student* | | | |
| Baseline | 35.44 | 34.43 | 49.81 |
| + DARGO-MI | 45.88 (↑10.44) | 44.81 (↑10.38) | 54.18 (↑4.37) |

Table 6: Ablation study of exploration in DARGO on BIRD-SQL.

| METHOD | BIRD-SQL (MI) |
|---|---|
| DARGO w/ Phi-3-mini | **33.80** |
| *(a) w/o MOI* | 21.70 (↓12.10) |
| *(b) w/o Case Study Trans.* | 22.80 (↓11.00) |
| *(c) w/o Functional Plan* | 23.60 (↓10.20) |

**Out-of-Distribution Evaluation of DARGO** Current tabular-reasoning benchmarks frequently provide *test-only* splits due to large costs of more complex tasks annotation, prompting us to evaluate how well DARGO generalizes when *no* in-domain training data are available. Following the [55], we first construct a unified memory from our exploration corpus and evaluate its performance on those *test-only* sets via DARGO. To be specific, exploration set $\mathcal{B}$ contains: WikiTQ [41], TabMWP [34], BIRD-Pandas [29], and Juice [3]. The distributions of $\mathcal{B}$ are shown in Appendix B.4. The *test-only* datasets form the set $\mathcal{E}$, which includes CRT-QA [70], QR-DATA [32], and Infi-Agent [20].

During evaluation, when we test DARGO on benchmark $\mathcal{E}_i \in \mathcal{E}$, its corresponding exploration corpus is $\mathcal{B} \cup \mathcal{E}_{\setminus i}$. This strict partition guarantees zero overlap between examples used for distillation and those reserved for evaluation. As summarized in Table 5, DARGO presents huge OOD improvements across all student models, demonstrating that it distills transferrable reasoning patterns rather than memorizing dataset-specific artifacts.

### 3.5 Fine-Tuning v.s. DARGO Knowledge Distillation

**With the Same Seed Data.** We compare DARGO with LoRA fine-tuning [19] using the same 1,000 training samples on BIRD-SQL for Phi-3-mini. Figure 5 and Table 9 in Appendix shows that fine-tuning on limited data even degrades SLM performance, while DARGO achieves significant improvements. We attribute this to two factors: 1) small training sets introduce bias that limits generalization; and 2) LoRA struggles to teach complex reasoning capabilities in an end-to-end regime. In contrast, DARGO leverages LLMs to decompose difficult questions into interpretable steps and distill planning knowledge, enabling better generalization. This makes DARGO particularly effective for DACG domains with limited annotations.

**Comparison with Incremental Lora Fine-Tuning.** For a more comprehensive comparison, we conduct additional experiments fine-tuning Phi-3-mini on varying proportions (10%, 20%, 50%, 70%, 100%) of the full BIRD-SQL training set. Each configuration is repeated 5 times, with average results presented in Figure 5. Our findings reveal several notable patterns: First, performance peaks at 70% of the data, suggesting that fine-tuning on more data does not always lead to better results, which maybe due to the inclusion of lower-quality or biased examples in the full dataset. Second, while fine-tuning with 70-100% of the data marginally outperforms DARGO, **it requires 7-9× more training examples** to achieve this performance. Notably, DARGO outperforms fine-tuning on 50% of the data, demonstrating **5× greater data efficiency**.

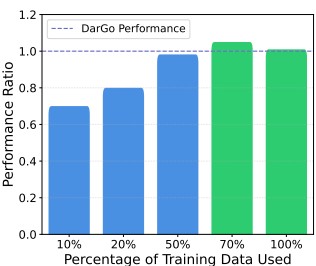

Figure 5: Comparison of DarGO with Lora fine-tuning across varying data size.

### 3.6 Ablation Study

In Table 6, to better validate the function of each component of DARGO, ablation studies are performed in both Exploration and Inference (More details in Appendix E):

**(a) MOI.** For the "w/o MOI" scenario, we construct the memory by employing GPT-4o to convert ground-truth code to plans directly, excluding the participation of SLM. This resulted in a significant final performance drop for Phi-3-mini. One explanation is that GPT-4o tends to produce relatively coarse-grained plans (e.g., `"compute percentage of winning"`), whereas SLMs require

more fine-grained steps (e.g., `"compute total number of games"`, `"divide winning games by total wins"`). These higher-level plans can induce hallucinations to SLMs, as they frequently entail additional intermediate steps that increase the risk of errors. Furthermore, without execution-based refinement and calibration through the Plan Optimization (PO) component in MOI, plans generated by GPT-4o often contain inaccuracies that can mislead the SLM, as illustrated in Section 2.2.

**(a) Case Study Translation.** As elaborated in Section 2.3, case study translation systematically consolidates and refines heterogeneous inputs, including tabular data, user queries, and code samples, into a structured, context-rich representation that SLMs can more effectively process. In the absence of this translation component, the model frequently defaults to errorous or incomplete outputs (e.g., "`SELECT \n\n\n\n...`"), underscoring the difficulty SLMs encounter in performing generalized reasoning when confronted with raw, unprocessed data in DACG.

**(c) Functional Plan.** A key contribution of our work is the MOI and the use of functional plans to structure and improve the interaction between teacher and student. We evaluate performance by replacing functional plans with textual plans like Chain-of-Thought. The result can prove that our designed functional plans are better orchestration media type compared to general COT plans in data science code generation task.

## 4   Related Work

**Data Analysis Code Generation (DACG).** DACG automates code generation for data-centric tasks in formats such as CSV, TSV, and relational databases (RDB) [7, 35]. It requires code that accurately handles schemas, formats, and data semantics, whether in Python for tabular data [9, 10, 47] or SQL for databases [68, 27, 29]. Spreadsheet-based code generation further extends DACG to formula generation in tools [54, 5]. Although large language models have shown promise, privacy remains a challenge in cloud-based environments [35].

**Knowledge Distillation.** Knowledge distillation can mitigate this problem by transferring LLM capabilities to smaller models, enabling efficient deployment in resource-constrained environments [63]. The field has evolved from early work on softened output training [18] to advanced techniques like task-specific fine-tuning [45], zero-shot learning [57], and instruction-following datasets [58, 57]. Progressive distillation techniques, such as the Orca framework [38], demonstrate the potential for guiding the development of efficient open-source models. Self-distillation approaches have explored autonomous training data generation [58]. Recent advancements have focused on improving the performance and privacy aspects of DACG by knowledge distillation [35]. At the same time, synthetic data has been leveraged to enhance the generalization of SQL generation across different schemas [64]. Even though these techniques are effective, most still require training efforts to transfer knowledge. Our DARGO framework introduces distillation through in-context learning, eliminating the need for task-specific fine-tuning.

**Memory for Large Language Models.** Memory can enhance LLMs by retaining long-term context and knowledge [69], as in reflection-based frameworks [48], subroutine reuse [50], and self-correction [4]. Complex tasks may require repository-level memory [65, 56, 55]. Typically, memory remains confined to a single model. Our work introduces multi-model memory orchestration (via GPT-4o), enabling smaller models to tap into broader knowledge sources for DACG.

## 5   Conclusion

In this paper, we presented DARGO, an automatic framework for knowledge distillation from Large Language Models to Small Language Models in Data Science Code Generation. DARGO leverages In-Context Learning to enhance SLM performance without fine-tuning, using model orchestration and memory-based distillation to improve task accuracy. Evaluations on three challenging tabular data analysis datasets that require code generation show a 27.5% relative performance increase for Phi-3-mini. We also show model-agnostic effectiveness, benefiting other SLMS, even they did not participate in the orchestration. These results highlight the potential of DARGO for developing intelligent applications with a focus on privacy and computational efficiency.

# 6 Acknowledgments

We thank the anonymous reviewers and committees for their helpful comments, suggestions and organizations. We appreciate Xinnuo Xu, Christian Poelitz, Siân Lindley, Hank Lee, Xiaolong Li, Ge Qu for discussion and suggestions. Reynold Cheng, Jinyang Li, were supported by the Research Grant Council of Hong Kong (RGC Project HKU 17202325), the University of Hong Kong (Project 2409100399), and the HKU Faculty Exchange Award 2024 (Faculty of Engineering).

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

# A  Model Implementation

We implement models for three main categories of purpose:

## A.1  Orchestration Models

`gpt-4o`: The Teacher model (`gpt-4o`) is responsible for several key tasks, including Abstraction Lifting (see Section 2.2) and Plan Optimization (see Section 2.2), which are performed while monitoring the performance of the Student model. Additionally, the Teacher model handles the conversion of complex, heterogeneous cases into more readable case studies for Student Learning Models (SLMs), as detailed in Section 2.3.

`Llama-3.3-70B-Instruct`: we use this powerful open-source model as teacher to prove the generalization of DARGO workflow.

`phi-3-mini-128k-instruct`: For the orchestration process, we select this 3.8B parameter SLM as the Student model due to its strong generalization abilities and efficient deployment.

`llama-3.1-8b-instruct`

## A.2  Baseline Models

Within the orchestration mode, several families of Student Learning Models (SLMs) are evaluated. These include models from the Phi-3, Starcoder 2, and LlDARGO families:

### Phi-3 Family   [1]

`phi-3-mini-128k-instruct` (3.8B)

`phi-3-small-128k-instruct` (7B)

`phi-3-medium-128k-instruct` (14B)

### Starcoder 2 Family   [33]

`starcoder2-7b-instruct`

`starcoder2-15b-instruct`

### Llama Family   [13]

`codellama-7b-instruct-hf`

`codellama-13b-instruct-hf`

## A.3  Models in Knowledge Transmission

In Section 3.4, we explore the knowledge distilled from DARGO to newly developed models, particularly in terms of their ability to generalize knowledge. For this evaluation, we select the following models:

`llama-3.1-8b-instruct`: This model is broad new, yet it shows significant performance improvements when leveraging the distilled knowledge.

`gpt-35-turbo-16k`: We also include a closed-source model in our experiments to demonstrate the effectiveness of our approach across both GPU-deployed and API-based models. Despite its number of parameters is unknown, we consider it as one of SLMs since its performance falls behind of its more advanced versions such as GPT-4.

## A.4  Demonstration V.S. Distillation

Given the memory database, we compare the effectiveness of our knowledge distillation techniques, with conventional demonstration-based strategies. In our approach, distillation involves transferring knowledge from the memory database to SLMs through task-specific instructions. On the other hand,

demonstration-based methods guide SLMs by presenting explicit task examples to facilitate analog reasoning [67]. We implemented two variants of few-shot demonstrations: **Static**: Human experts select three representative examples from the memory database, which remain constant across all cases. **Dynamic RAG-based**: Examples are selected from memory database based on similarity to the current query. For fair comparison, we also implement the same RAG system as DarGO-MI, described in Section 3.2.

Our findings show that few-shot demonstration generally underperforms DarGO across datasets. However, RAG-based few-shot demonstration outperforms both our designed knowledge distillation and other baselines on TABMWP. This success appears linked to the simplicity of TABMWP's input data, which averages 2.22 columns and 6.13 rows per data point, with clean numeric or processed string values (Table 7). In contrast, for WIKITQ with irregular value types, and BIRD-SQL with complex schemas, SLMs struggle, generating 38.2% more invalid outputs, such as erroneous SQL queries, in BIRD-SQL. Based on these observations, we conclude that dynamic few-shot demonstration is more convenient and effective for leveraging the memory database when the input data is less complex. On the contrary, for complex data such as tables with dirty values or relational databases, our designed knowledge distillation enables SLMs to better utilize knowledge and perform tasks more effectively.

# B  Dataset Implementation Details

## B.1  Train-Test Distillation Data Statistics

Table 7 summarizes the distributions of the datasets used in this study. These datasets include both single-table and multi-table relational structures, offering a comprehensive aspects for evaluation. The question types range from standard semantic parsing queries to more complex analytical questions involving aggregations and nested operations. This diversity enables a thorough assessment of the model's performance across various query paradigms.

## B.2  Data File Content

For convenient reproduction and following, we preprocess all dataset into more unified data format of `jsonl`. In python task (TABMWP, WIKITQ), each line of data contains `question_id`, `question`, `data_path`, `data_overview`, `answer_type`, `answer`. In SQL task (BIRD-SQL), each line of data contains `question_id`, `question`, `evidence`, `data_path`, `db_id`, `sql`.

## B.3  Data Input Content

The main goal of this work is to evaluate the code generation capabilities of models in understanding data schemas and structures across

Table 7: Statistics for three datasets. The term `Analysis` indicates that the dataset mainly consists of analytical questions, while `SP` refers to semantic parsing tasks.

| STATISTIC | TABMWP | WIKITQ | BIRD-SQL |
|---|---|---|---|
| **Dataset Features** | | | |
| # train examples | 1,000 | 2,000 | 1,000 |
| # eval examples | 1,000 | 1,000 | 500 |
| question type | Analysis | SP | SP + Analysis |
| # toks / Q | 26.5 | 12.6 | 20.0 |
| **Data Structure** | | | |
| data input type | Single | Single | RDB |
| # rows / data | 6.13 | 28.5 | 354k |
| # columns / data | 2.22 | 6.36 | 73.3 |
| **Code Features** | | | |
| code type | Python | Python | SQL |
| answer type | String | String | Code |
| # toks / code | N/A | N/A | 61.15 |

multiple datasets. Given the impracticality of providing all data values in real-world scenarios in which datasets may consist of millions or even billions of rows, we sample values for the part of data input to simulate realistic code generation tasks. We feed the markdown format of schemas with data samples as `data_overview`.

For **TABMWP**, we provide only the column names and the first three rows of values. This enables models to infer the data structure and value types necessary for Python Pandas code generation without exposing all the data.

For **WIKITQ**, which contains more complex and varied value types, we provide the first 10 rows of values and column names to help models navigate the dataset's intricacies.

In the case of **BIRD-SQL**, which contains relational databases with complex schemas and diverse value types, more advanced schema-linking techniques are often required to retrieve relevant tables

or columns before answering queries [53, 43]. While we consider this advanced schema-linking process as future work for DARGO, our current focus is on the code generation aspect. Therefore, we provide:

- Ground truth retrieved tables, reducing input complexity and simulating realistic human-machine interactions where users might supply potentially relevant tables.
- Full columns with column meaning description files.
- The first three rows of values for each table.

Although the retrieved tables are given, the models must still consider constraints and generate correct SQL queries. As shown in Table 2, performance on Bird-SQL remains relatively low, even with simplified table retrieval, highlighting the challenges of generating accurate SQL queries in complex database environments. This methodology allows us to evaluate code generation capabilities while approximating the real-world challenges of data analysis.

### B.4 OOD Cross-Dataset Evaluation Set Statistics.

Table 8 show the data statistics of basic set of unified memory for evaluating DARGO on OOD cross-dataset distillation.

Table 8: Statistics for Basic Set of Unified Memory.

| Dataset | # Items |
|---------|---------|
| wikitq | 1 000 |
| TabMwp | 1 000 |
| BIRD-Pandas | 300 |
| Juice | 1 000 |

## C   Action Types

**Decomposition.**   The `Decomposition` action type divides a large, multifaceted step `x` into multiple simpler steps `a` and `b`. By breaking complex workflows into smaller components, the Model Orchestration Interface (MOI) ensures that the resulting plan is both clearly understood and more straightforward for an SLM to execute. This approach clarifies the logic behind each sub-step and allows finer-grained control over how tasks are performed or combined, thereby reducing confusion and facilitating future refinements.

**ALT.**   Sometimes, the instructions for a step can be ambiguous, incorrect, or unnecessarily complex. The `ALT` action type replaces such a problematic step `x` with a newly clarified step `y`. By substituting erroneous or unclear instructions, the MOI ensures that the plan adheres to more accurate logic, minimizing the likelihood of misinterpretation and promoting consistency throughout the workflow.

**ADD.**   When a critical operation is missing or an additional step is required for completeness, the `ADD` action type introduces a new step `a` into the existing plan. Adding steps proves valuable when the plan overlooks essential checks, transformations, or other auxiliary procedures. This mechanism ensures more thorough and dependable solutions.

**DELETE.**   Certain steps can be redundant or risk causing confusion for subsequent code generation. The `DELETE` action type removes any unnecessary step `x`, thereby streamlining the plan. By eliminating irrelevant instructions, the MOI reduces cognitive load on the SLM and maintains a logically consistent sequence of steps that aligns directly with the overarching goal.

**SIMPLIFY.**   Whenever possible, it can be advantageous to simplify complex steps. The `SIMPLIFY` action type replaces a complicated step `x` with a more direct version, `simple_step(x)`. For example, it may transform a solution relying on recursion into an iterative, loop-based approach. Simplification improves both computational efficiency and interpretability, since SLMs often perform better with concise instructions.

Table 9: Performance evaluation of Zero-Shot End-to-End Code Generation, LoRA fine-tuning, and our proposed knowledge distillation techniques on BIRD-SQL. Deeper red shading indicates a larger performance drop compared to the original pre-trained model, while green indicates no decline or improvement.

| Model | SIMPLE | MEDIUM | CHALLENGING | OVERALL |
|---|---|---|---|---|
| *Zero-Shot End-to-End Code Gen.* | | | | |
| Original Checkpoint | 38.51 | 21.20 | 11.76 | 24.40 |
| *LoRA Fine-Tuned* | 39.86 | 19.20 | 10.78 | 23.60 |
| DARGO *Knowledge Distillation* | | | | |
| Meta Instruction | **51.35** | **30.40** | 16.67 | **33.80** |

**SWITCH.**   Selecting an appropriate library or tool is critical for achieving both correctness and simplicity. The `SWITCH` action type replaces a step implemented with `packageA` by an equivalent (or more suitable) operation from `packageB`. For instance, one might switch from using `LinearRegression` to `CorrelationCoefficient` when the task is simply to determine the relationship between two variables. This approach avoids unnecessary overhead and preserves clarity in the plan.

By employing these six action types, `Decomposition`, `ALT`, `ADD`, `DELETE`, `SIMPLIFY`, and `SWITCH`, the MOI systematically refines high-level plans. This process results in efficient and easily interpretable workflows, ensuring consistency from the design of the plan to its final implementation by SLMs.

# D   DARGO Functionality

## D.1   MOI Generalization

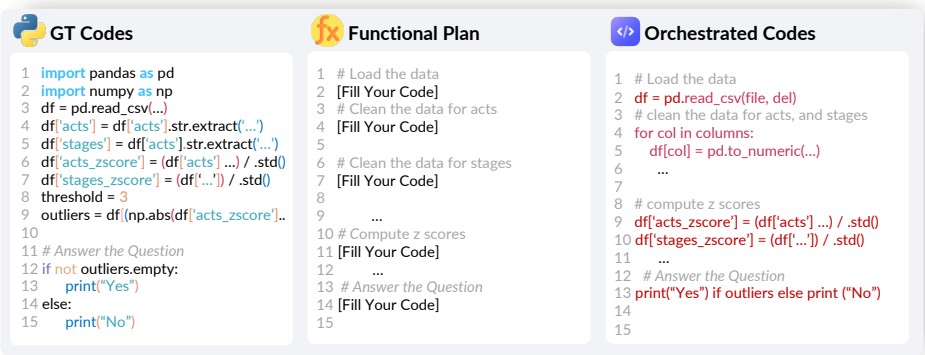

Figure 6: Illustration of how MOI is conducted in Python for Tabular data analysis.

Our Model Orchestration Interface (MOI) is adaptable to different programming languages with different data input settings. Figure 2 shows how MOI is conducted in RDB settings with SQLite, and Figure 6 shows how it's undertaken in Single-tabular data with Python.

## D.2   Fine-Tuning v.s. DARGO Knowledge Distillation

# E   Ablation Study for Inference

We conducted a comprehensive ablation study of DARGO-MI, as shown in Table 10. Code-T5+ is a code embedding model [59], while BGE-Large [61] represents one of the state-of-the-art (SOTA) text embedding models. The study examines two types of RAG Index: one where distance is computed using question embeddings alone, and another where both question and schema embeddings are used. The "Plan + Gen" approach involves first constructing a plan with distilled knowledge, followed by generation using knowledge-driven planning. In contrast, the "Gen" approach involves direct

generation without prior planning. The instruction type labeled `w/ examples` refers to cases where a specific example is provided by the Teacher model. We evaluate performance with 1, 3, and 5 examples to assess the impact of varying numbers of RAG examples. The results of the ablation study reveal several key insights:

**Code embeddings outperform text embeddings.** The superior performance of Code-T5+ over BGE-Large-en can be attributed to the nature of the task. While text embeddings emphasize on semantic and domain knowledge, code embeddings capture the syntactic and logical structure of coding problems, which is crucial for DACG tasks. Even when presented with identical questions, the code solutions can vary significantly depending on the data input. Code-T5+ is able to effectively embed questions from a programming perspective, benefiting from its pre-trained corpus, whereas text embeddings are less suited for the task.

**Embedding only the question is more effective than embedding both the question and schema.** The study demonstrates that question-only embeddings lead to better results. This suggests that the inclusion of schema in the embedding may introduce unnecessary complexity, which may hinder performance on the DACG task.

**Planning is essential for more complex tasks.** The results stress on the importance of planning in a knowledge-driven generation. For tasks requiring complex reasoning, the "Plan + Gen" approach outperforms direct generation (`Gen`), indicating that structured planning significantly improves task performance.

**One example may bias the SLM.** Involving a single example in the instruction can introduce bias in Small Lanuguage Models (SLMs). A specific example might cause the SLM to over-follow to certain information, leading to hallucinations. For instance, if the example includes a reference to `"singer"`, the SLM may generate plans that include `"singer"` even when the question pertains to an unrelated topic, such as `"cars"`. This observation highlights the lack of robustness in SLMs when exposed to overly specific examples. Consequently, it is better to provide more general, transferable knowledge in instructions. The degraded performance observed with 1 RAG example supports this conclusion, as the model becomes overly reliant on the provided information.

**More examples do not always improve performance.** Interestingly, increasing the number of RAG examples (from 1 to 5) results in a performance drop. This suggests that longer input sequences may confuse the SLM, making it more difficult to distill relevant knowledge. Based on these findings, we recommend using 3 RAG examples as the optimal balance for complex DACG tasks since it avoids both the biases of a single example and the confusion caused by too many examples.

Table 10: Ablation Study Results of DARGO-MI of Phi-3-mini on BIRD-SQL. The table compares different embedding models, RAG index (with or without schema), reasoning approaches (planning or direct generation), and varying numbers of RAG examples.

| Embedding Model | RAG Index | Reasoning Type | Instruction Type | # RAG Examples | Performance |
|---|---|---|---|---|---|
| code-t5+ | question | plan + gen | no examples | 3 | 33.80 |
| code-t5+ | question | gen | no examples | 3 | 31.40 ($\downarrow$2.40) |
| bge-large | question | plan + gen | no examples | 3 | 30.00 ($\downarrow$3.80) |
| code-t5+ | question | plan + gen | w/ examples | 3 | 28.00 ($\downarrow$5.80) |
| code-t5+ | question+schema | plan + gen | no examples | 3 | 32.40 ($\downarrow$1.40) |
| code-t5+ | question | plan + gen | no examples | 5 | 31.80 ($\downarrow$2.00) |
| code-t5+ | question | plan + gen | no examples | 1 | 29.80 ($\downarrow$4.00) |

# F  Error Analysis

We conducted an error analysis by sampling 50 incorrect cases for both DARGO-MI across three datasets. Although DARGO substantially improves the overall performance of SLMs, we found that 54% of the errors were caused by over-reasoning. This issue tends to emerge even in relatively simple cases. As discussed earlier, SLMs can overly adhere to the instructions derived from planning and guidance, which is problematic when the task is enough simple and does not require decomposition or reasoning. In these cases, direct code generation would lead to more accurate results. The remaining

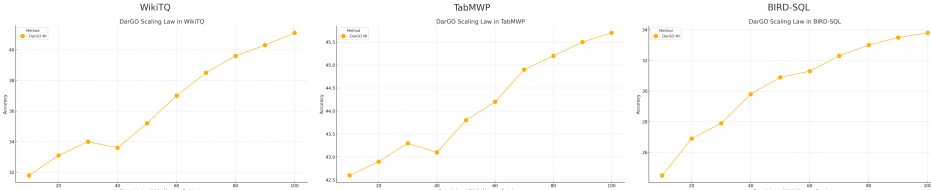

Figure 7: The scaling performance of DARGO on three main DACG datasets.

errors stem from common issues in code generation tasks, such as incorrect string handling, incorrect column selection, database constrain understanding.

# G    Limitations and Future Work

A key limitation of our current approach with DARGO is the reliance on initial training examples for both LLMs and SLMs to facilitate orchestration. This is why we selected datasets that include a training corpus suitable for distilling knowledge. However, an important avenue for future work is to explore how to generate such training data in a fully zero-shot manner, without relying on human-annotated or enumerated examples. Additionally, as highlighted in the error analysis, over-reasoning negatively impacts performance on simpler tasks, where additional reasoning or decomposition is unnecessary. To address this, future work could focus on developing or prompting smaller models to act as routers, as proposed by Ding et al. [12], to classify questions based on whether they require planning. This would help avoid over-reasoning in straightforward cases and improve the overall efficiency of DARGO.

# H    Scaling Analysis of DARGO

Figure 7 shows that DARGO scales reliably with increasing data volume. Although its performance is somewhat unstable on very small memory, stability improves rapidly, and accuracy surpasses the baseline once more than 60% of the exploration data are available which proves that the distilled knowledge is being exploited effectively.

# I    Cost Analysis

We compare the cost of DARGO with other distillation methods shown in Table 12, Table 16 and Table 14. And Table 11, Table 15, Table 13 show the inference cost of DARGO and its comparison with other baselines. We can conclude that: first, it shows DarGO now is the most efficient Distillation work for this task. Second, since inference is conducted by SLMs, the additional token usage is acceptable when considering its significant improvement.

Table 11: BIRD-SQL: Infer (Tab. 16)

| Method | SLM InToks ↓ | SLM OutToks ↓ | EX ↑ |
|---|---|---|---|
| End2End (baseline) | 358,277 | 16,751 | 24.40 |
| Chain-Of-Thought | 369,433 | 19,317 | 23.80 |
| Static Few-Shot | 794,342 | 17,306 | 26.20 |
| Dynamic Few-Shot | 836,678 | 26,291 | 24.40 |
| DSPy | 637,146 | 60,748 | 22.80 |
| ReGAL | 1,602,698 | 16,045 | 20.40 |
| DarGO-MI | 647,799 | 64,836 | 33.80 |

# J    Broader Impact

Given DARGO is designed for automating code generation for better and efficient data analysis, it can help data scientists explore the potential risks of financial market, earthquake, which are beneficial to

Table 12: BIRD-SQL: Exploration (Tab. 17)

| Method | LLM InToks | LLM OutToks ↓ | Cost ↓ |
|--------|------------|---------------|--------|
| DSPy | 18,129,504 | 2,689,076 | $108.32 |
| ReGAL | 13,259,759 | 2,381,368 | $85.44 |
| DarGO | 11,086,194 | 1,372,421 | $62.16 |

Table 13: WikiTQ: Infer (Tab. 18)

| Method | SLM InToks ↓ | SLM OutToks ↓ | Acc. ↑ |
|--------|--------------|---------------|--------|
| End2End (baseline) | 235,122 | 44,033 | 32.50 |
| Chain-Of-Thought | 242,443 | 149,246 | 27.70 |
| Static Few-Shot | 521,292 | 45,161 | 23.00 |
| Dynamic Few-Shot | 549,075 | 93,415 | 16.60 |
| DSPy | 418,131 | 123,422 | 26.70 |
| ReGAL | 2,082,898 | 162,949 | 36.10 |
| DarGO-MI | 495,493 | 168,311 | 41.10 |

the society. Also, our study only focuses on code generation, a programming-level language rather than natural language, it will not impact society negatively.

# K    Reproducibility

We provide codebase in the supplementary files and we list all implementation details in Appendix B And we deliver prompts for each stage and baselines in Appendix L for fully reproducibility.

# L    Main Prompts

The zero-shot End-to-End Code Generation prompt is shown in Figure 8, Figure 16 and 18 show the zero-shot Chain-Of-Thought reasoning. Figure 19 shows few-shot demonstration prompting. The `few_shot_examples` can be selected by human experts as Static Few-Shot Demonstration, and can be retrieved from DARGO memory database by RAG system as Dynamic Few-Shot Demonstration.

The Figure 8, 9, 10, 11 show prompts for Orchestration between LLMs and SLMs. Figure 12 presents how LLM convert orchestrated successful cases to more understandable case studies to SLMs. LLMs can go through correct cases from memory databases and distill knowledge to an offline and plug-and-plan General Instruction for SLMs to used for new and unseen queries performed by prompts shown in Figure 13 and 14. During inference, SLMs can produce Meta Instructions by prompts in Figure 15. Given distilled knowledge (instructions), SLMs will plan first as shown in Figure 17, and generate codes finally with their knowledge-driven planning, which shows in Figure 18.

# M    Knowledge Distillation Examples

## M.1    Case Study Example

The Figure 20 shows the example of case studies on Python task. Figure 22 present examples of DARGO-MI respectively.

Table 14: WikiTQ: Exploration (Tab. 19)

| Method | LLM InToks | LLM OutToks ↓ | Cost ↓ |
|--------|-----------:|--------------:|--------|
| DSPy   | 10,893,180 | 1,616,141     | $65.09 |
| ReGAL  | 7,967,176  | 1,431,207     | $51.34 |
| DarGO  | 6,661,181  | 824,828       | $37.35 |

Table 15: TabMWP: Infer (Tab. 20)

| Method | SLM InToks | SLM OutToks | Acc. Score |
|--------|-----------:|------------:|-----------:|
| End2End (baseline) | 171,480 | 38,016  | 41.80 |
| Chain-Of-Thought   | 176,764 | 39,244  | 41.40 |
| Static Few-Shot    | 380,177 | 38,283  | 36.20 |
| Dynamic Few-Shot   | 400,358 | 52,578  | 52.10 |
| DSPy               | 304,898 | 49,073  | 40.40 |
| ReGAL              | 767,114 | 37,679  | 41.80 |
| DarGO-MI           | 314,817 | 125,230 | 45.70 |

Table 16: TabMWP: Exploration (Tab. 21)

| Method | LLM InToks ↓ | LLM OutToks ↓ | Cost ↓ |
|--------|-------------:|--------------:|--------|
| DSPy   | 9,426,638    | 893,604       | $48.75 |
| ReGAL  | 7,790,084    | 751,736       | $40.49 |
| DarGO  | 5,330,507    | 659,862       | $29.89 |

```
You are a data analyst. Given the data, you need to generate the code first to answer the question:

# Please Follow:
- Do not add data inspection in the plan, such as `df.head()` or `print(df.head())` since this is
cheating!
- Do not use any external information.
- The code should be end-to-end, so you cannot encourage yourself to print other things except
  final result. More other information lead to be distracted.

# Question: {question}
# Thought: I need to see the data samples in the first 10 rows:

# Code:
```python
import pandas as pd
df = pd.read_csv('{data_path}', sep='\t')
print(df.head(10))
```

# Observation:
{data_overview}

# Thought: I can generate remaining code to answer this question:
# Code:
```python
import pandas as pd
```

Figure 8: Prompt of baseline end-to-end generation for tasks requiring Python.

```
You are a data analyst trainer. You are educating your student to generate right code to answer
tabular data analysis questions. In order to do so, you need to convert your code to code_plan and
let your students to fill to understand plans and analysis. So you cannot generate code by your
own, only generate plans.

# Data Overview at the path {data_path} (first ten rows):
{data_overview}
...

# Question: {question}
# Original Code:
```python
{ground_truth code}
```

You should convert the code into the code_plan format with the placeholder `[FILL YOUR CODE HERE]`:
```code_plan
import ...

# Step 1:....
[Fill Your Code]

# Step 2:....
[Fill Your Code]
...

# Step N: ....
```

Generate your code_plan for your student. DO NOT generate any code by your own. Also ignore and
remove steps of inspecting the data which leads to student cheating.
Please note it's hard for your student to write long code. You will get 1,000 dollars if you have
a good job:
```

Figure 9: Prompt converting ground-truth code to functional plan for python task as example. This is conducted by **LLM Teacher model**.

```
You are a data analyst. Given the data, expert customized functional plan, complete
each line of code to answer questions correctly:

# Please Follow:
- Do not add data inspection in the plan, such as `df.head()` or `print(df.head())`
since this is cheating!
- Do not use any external information.
- The code should be end-to-end, so you cannot encourage yourself to print other
  things except final result. More other information lead to be distracted.

# Question: {question}
# Function Plan:
```python
{functional plan}
```
# Your entire completion code for function plan executable and correct:

# Code:
```python
import pandas as pd
```

Figure 10: Prompt of orchestration coding. This is conducted by **SLM Student model**.

```
You are an expert in error analysis and code planning. Your task is to guide your intern in filling out the code for your logic. You need to generate textual plans
as comments that include essential import statements, logics. Currently, the mixed code filled by your intern is incorrect. Then you should analyze and help him.

------------------------------------------ case begin: ------------------------------------------------
{last turn case}
------------------------------------------ end ------------------------------------------

You are experienced data analysis programmer responsible for checking the errors, analyzing the reasons, and helping them correct the code. Note that you cannot
fill the code for them directly. You have four options for actions:
1. **Decomposition(Step Number, new sub steps**: If a step is too complicated and exceeds the intern's capability, decompose this step into multiple smaller steps
for them to fill step by step.
    Actually, you have to decompose steps if there are multiple functions or multiple lines of code in one step since they are not capable!
    step a -> step b, step c
2. **ALT(Step Number, what do you want to alt in details**: If a step is ambiguous or requires additional information or options, provide an alternative approach
or clarification. But this is a closed-book education, you cannot teach them to use external information aside code and data samples.
    step a -> step b
3. **ADD(Step Number, what do you want to add in details**: If the original step lacks important operations, add a supplementary step to ensure the main code logic
is smooth. But this is a closed-book education, you cannot teach them to use external information aside code and data samples.
    Also all available data are shown, you cannot add or teach them to use `df.head()` to overview data again.
    step a, step c -> step a, step b, step c
4. **DELETE(Step Number, what do you want to delete in details**: If some steps are unnecessary and hinder the intern's understanding of the overall logic, delete
them.
    step a, step b -> step b (deleted step a)
5. **SIMPLIFY(Step Number, simplify specific steps)**: If a step is implemented using recursion and this approach is too complex for the intern to understand or
debug, suggest a non-recursive approach that achieves the same result.
This might involve using iterative methods or other strategies to simplify the logic. If you find code fails due to this, simplify the functions.
step a (recursive) -> step a (iterative)
6. **SWITCH(Step Name, packages to SWITHC)**: If a function relies heavily on a specific package that is known to be complex or not beginner-friendly, suggest
switching to a more intuitive or simpler package that achieves similar functionality. This can help the intern understand the underlying logic without getting
bogged down by the complexities of the original package.
    step a (uses ComplexPackage) -> step a (uses SimplePackage)

You have to provide reasons based on analysis of errors for choosing this action and show your action in <action></action>, then. Finally, you must execute your
chosen action to change original code and fill in the following format:

# format:
Reason:
<reason>...</reason>
Act:
<action>...</action>

# Updated code plan:
```code_plan
import ...

# Step 1:....
[Fill Your Code]

# Step 2:....
[Fill Your Code]
...

# Step N: ....
```

## Please note:
- Do not ask students to add data inspection in the code, such as `df.head()` or `print(df.head())`
- Do not guide them to use any external information.
- The mixed code should be end-to-end, so you cannot encourage student to print other things except the final result. More other information would cause student to
be distracted.
- Just focus on how to make students learn how to better plan in the end-to-end code generation.

OK, now change your codes according to your actions.
If you don't follow rules, then you will lose 1 million dollars:
```

Figure 11: Prompt of plan optimization. This is conducted by **LLM Teacher model**.

```
You are a data analyst trainer. You are educating your student to generate pythyon code to answer tabular data analysis questions.

This is a successful case of your code, perform a case study on this:
------------------------------------------ case begin: ------------------------------------------------
# Question: {question}

# Data Overview at the path {data_path} (first 10 rows):
{data_overview}
...

# Code:
```python
{final orchestrated code}
```
------------------------------------------ case end: ------------------------------------------------

perform a concise case study! Your case study should only contain

### Case Study: [Case Name]
### Question: [Question]
### Table Info: [Summarized Useful information about Tabular Data]
### Objective:
### Explanation:

Please note your case study should make your student understand. You don't have to include code again. You will get 1000 dollars if
you have a good job:
```

Figure 12: Prompt of case study conversion. This is conducted by **LLM Teacher model**.

```
You are a data analyst trainer. You are educating your student to generate correct python pandas code to answer tabular
data analysis questions. To test and elicit their knowledge of python pandas code, you generate step-by-step plans that
allow them to fill in code until they succeed.

# These are case studies where they fill the correct code:
------------------------------------------ case begin: ------------------------------------------------
{case_study_batch}
------------------------------------------ case end: ------------------------------------------------

# Please note:
- Do not ask students to add data inspection in the code, such as `df.head()` or `print(df.head())`
- Do not guide them to use any external information.
- The mixed code should be end-to-end, so you cannot encourage student to print other things except the final result. More
other information would cause student to be distracted.
- Just focus on how to make students learn how to better plan in the end-to-end code generation.

According to the previous case studies, analyze and reflect how to generate plans which can make your student fill the
correct code. Summarize 5-7 key points.
```

Figure 13: Prompt of aggregation prompt of each batch of case studies. This is conducted by **LLM Teacher model**.

```
You are a data analyst trainer. You are educating your student to generate correct python pandas code to answer tabular
data analysis questions. To test and elicit their knowledge of code, you generate step-by-step plans that allow them to
fill in subqueries until they succeed.

# These are case studies where they fill the correct code:
----------------------------------------- case begin: -----------------------------------------
{last layer of case studies}
----------------------------------------- case end: -----------------------------------------

# Please note:
- Do not ask students to add data inspection in the code, such as `df.head()` or `print(df.head())`
- Do not guide them to use any external information.
- The mixed code should be end-to-end, so you cannot encourage student to print other things except the final result. More
  other information would cause student to be distracted.
- Just focus on how to make students learn how to better plan in the end-to-end code generation.

Following case studies, please summarize 5-7 key points about how to plan and generate correct code to answer the tabular
data analysis questions accurately.
Students will take your notes directly.

# You need to start with:
```1. You should
```

Figure 14: Prompt of summarization prompt of batch of case studies in the last layer. This is conducted by **LLM Teacher model**.

```
You are a data analysis trainer. Your are teaching your student to plan and generate python code
accurately. You find some case study for reference.

# There are case studies:
----------------------------------------- case begin: -----------------------------------------
{case_studies}
----------------------------------------- case end: -----------------------------------------

Following case studies, please summarize key 5-7 points about how to plan and generate correct python
code to answer the data analysis questions accurately.

# You will use them to educate your student:
```successful plan suggestions:
1. You Should
```

Figure 15: Prompt of in-time summarization for meta-instructions. This is conducted by **SLM Student model**.

```
You are a data engineer. Given the sample data, generate python code plan to answer the question
accurately.

# Please Follow:
- Do not add data inspection in the plan, such as `df.head()` or `print(df.head())` since this is
cheating!
- Do not use any external information.
- The code should be end-to-end, so you cannot encourage yourself to print other things except final
  result. More other information lead to be distracted.
```

# Question: {question}
# Thought: I need to see the data samples in the first 10 rows:

# Code:
```python
import pandas as pd
df = pd.read_csv('{data_path}', sep='\t')
print(df.head(10))
```

# Observation:
{data_overview}

# Thought: I should have a step-by-step text plan for generating this code first. I will fill my plan
into the template in details:
```code_plan
Step 1: ...
Step 2: ...
...
Final Step: ...
```

Generate your plan step by step for the question:

# Let's think step by step:
```code_plan
Step 1:
```

Figure 16: Prompt of generating Chain-Of-Thought. This is conducted by **SLM Student model**.

```
You are a data engineer. Given the sample data, generate python code plan to answer the question

# Please Follow:
- Do not add data inspection in the plan, such as `df.head()` or `print(df.head())` since this is
cheating!
- Do not use any external information.
- The code should be end-to-end, so you cannot encourage yourself to print other things except final
  result. More other information lead to be distracted.

# There are some important successful plan suggestions from experts:

```successful plan suggestions:
{successful_plan_suggestions}
```

# Question: {question}
# Thought: I need to see the data samples in the first 10 rows:

# Code:
```python
import pandas as pd
df = pd.read_csv('{data_path}', sep='\t')
print(df.head(10))
```

# Observation:
{data_overview}

# Thought: Referring to [successful plan suggestions], I should have a step-by-step text plan for
generating this code first. I will fill my plan into the template in details:
```code_plan
Step 1: ...
Step 2: ...
...
Final Step: ...
```

Generate your plan step by step for the question:

# Let's think step by step:
```code_plan
Step 1:
```

Figure 17: Prompt of knowledge-driven planning. This is conducted by **SLM Student model**.

```
You are a data engineer. Given the sample data, generate python code to answer the question accurately.

# Question: {question}
# Thought: I need to see the data samples in the first 10 rows:

# Code:
```python
import pandas as pd
df = pd.read_csv('{data_path}', sep='\t')
print(df.head(10))
```

# Observation:
{data_overview}

# Thought: I can generate code to answer this question and print the result. I will fill my code in the
template:
```python
[Your Code]
```

Let's think step by step for the question:
{step-wise plans}

# Code:
```python
import pandas as pd
```

Figure 18: Prompt of code generation given step-wise planning. This is conducted by **SLM Student model**.

```
You are a data analyst. Given data sample, you need to generate pandas code first to answer the question.

Generate your pandas code to answer the question, and print the result for your to understand. Fill your
code in
```python
[Your Code]
```

# Please follow:
- Do not add data inspection in the plan, such as `df.head()` or `print(df.head())` since this is
cheating!
- Do not use any external information.
- The code should be end-to-end, so you cannot encourage yourself to print other things except the final
result. More other information would cause sutdent to be distracted.

There are some examples:
-------------------------- Examples Start --------------------------
{few_shot_examples}
-------------------------- Examples END --------------------------

# Question: {question}
# Thought: I need to see the data samples in the first 10 rows:

# Code:
```python
import pandas as pd
df = pd.read_csv('{data_path}', sep='\t')
print(df.head(10))
```

# Observation:
{data_overview}

# Thought: I can generate code to answer this question:

# Code:
```python
import pandas as pd
```

Figure 19: Prompt of few-shot demonstration. This is conducted by **SLM Student model**.

```
### Case Study: Average Weight Calculation for Specific Players

### Question:
What is the average weight of Jamarr Sanders and Robert Williams?

### Table Info:
- **Columns**: Name, Height, Weight (lbs.), Position, Class, Hometown, Previous Team(s)
- **Sample Data**:
  - Jamarr Sanders: Weight 210 lbs.
  - Robert Williams: Weight 210 lbs.

### Objective:
To calculate the average weight of the players Jamarr Sanders and Robert Williams from the given dataset.

### Explanation:
1. **Load Data**: The data is loaded from a tab-separated values (TSV) file.
2. **Filter Data**: Rows corresponding to the names "Jamarr Sanders" and "Robert Williams" are filtered from the dataset.
3. **Calculate Average**: The average weight of the filtered rows is computed.
4. **Output**: The result is printed as an integer.

By following these steps, the student can understand how to filter specific rows in a dataset and perform calculations on the filtered data. This case
demonstrates the practical application of data manipulation and analysis using pandas in Python.
```

Figure 20: Example of case studies for tasks requiring Python. This is conducted by **LLM Teacher model**.

```
1. You should **break down the task into manageable steps**. Each step should build on the previous one, guiding you through the process
logically.

2. You should **emphasize data handling and cleaning**. This includes handling missing values, normalizing case, and ensuring data consistency."

3. You should **focus on filtering and extraction**. Guide yourself on how to filter and extract relevant data based on specific criteria. This is often
the core of the analysis."

4. You should **perform aggregation and counting**. Learn how to perform aggregation operations like counting, summing, or finding
minimum/maximum values to derive insights from the data."

5. You should **present the result clearly**. Ensure that the final step involves presenting the result in a clear and concise manner. This reinforces
the importance of communicating findings effectively."

6. You should **avoid distractions**. Keep the instructions focused on the end-to-end process without encouraging unnecessary intermediate
outputs or external information. This helps maintain your focus on the task at hand."""
```

Figure 21: Example of General Instruction for tasks requiring Python. This is conducted by **LLM Teacher model**.

> **"question": "which country rank last?"**
>
> 1. Understand the problem statement and the data structure.
>
> 2. Load the data using appropriate libraries (e.g., pandas).
>
> 3. Perform necessary data manipulation and cleaning.
>
> 4. Identify the relevant columns and values for the analysis.
>
> 5. Use appropriate functions and methods to **filter**, **sort**, and extract the required information.
>
> 6. Output the result in a clear and concise manner.

Figure 22: Example of General Instruction for tasks requiring Python. This is conducted by **SLM Student model** in time.

