# OpenReview forum: "Unlocking SLM Potential for Data Analysis Code Generation via Non-Parametric Knowledge Distillation"
_NeurIPS.cc/2025/Conference — NeurIPS 2025 poster_

### Official Review · Reviewer_Y9rz · 2025-06-15

**Clarity:** 3
**Significance:** 2
**Originality:** 2
**Rating:** 3
**Confidence:** 4

**Summary:**

This work asks: Can we transfer an LLM’s reasoning ability to a smaller, locally deployable student language model (SLM) without any parameter updates? To address this, the authors propose a non-parametric knowledge distillation framework that requires no fine-tuning, delivering LLM-level DACG capabilities on compact SLMs.
It introduces DARGO, a three-stage pipeline including abstraction lifting, orchestration coding, and plan optimization.
DARGO is evaluated on three challenging benchmarks, WIKITQ, TABMWP, and BIRD-SQL.

**Questions:**

1. How do you define small language model and large language model? How to distinguish them?
2. Which BIRD setting are you using? Is `Oracle Knowledge` provided to the model? This is a huge difference.

**Ethical Concerns:**

["NO or VERY MINOR ethics concerns only"]

**Final Justification:**

I will keep my original score. The authors’ initial response was somewhat evasive, offering common-sense statements rather than clarifying their claims in the paper. Moreover, the overall idea of this work is not novel, and the use of an oracle setting on BIRD in the experiments makes it mediocre rather than demonstrating strong reasoning and generalization capabilities.

**Limitations:**

1.  Repeated teacher calls is still not cheap. These multi-round executions may be impractical in low-budget or real-time settings.
2.  Large memory banks could slow down the retrieval part.

**Paper Formatting Concerns:**

1. In line 10, `knowledge` not `knoweldge`
2. The abbreviation “CoT” appears inconsistently; standardize “Chain-Of-Thought (CoT)” throughout.
3. `GPT-3.5-turbo` or `GPT-35-turbo`?

**Quality:**

3

**Strengths And Weaknesses:**

Strengths
1. Proposed a non-parametric, in-context distillation from LLM to SLM, obviating costly fine-tuning.
2. Covers planning (MOI), bank construction, and inference (RAG-MI), with thorough ablation studies validating each module.
3. Large improvements on multiple benchmarks using small models.

Weaknesses
1. The MOI phase demands multiple teacher-model calls and code executions, which may be expensive.
2. As task volume grows, retrieval latency and storage demands could become bottlenecks.
3. Focuses exclusively on Python/SQL for tabular data tasks, while other languages or code-generation scenarios remain unexplored.
4. Relying on closed-source LLMs as teachers still remain privacy and access concerns.

---

> ### Author Rebuttal · Authors · 2025-07-30
>
> > Concern 1: The MOI phase demands multiple teacher-model calls and code executions, which may be expensive.
>
> **A1:** Thanks for this concern. We think our method provides a cost-effective knowledge distillation pipeline. We computed a detailed cost analysis in Appendix I with Table 12, Table 19, and Table 21. We show that our method is cheaper compared to popular knowledge distillation approaches DSPy (for general NLP tasks) and ReGAL (specifically for code tasks) whilst providing better performance.
>
> > Concern 2: As task volume grows, retrieval latency and storage demands could become bottlenecks.
>
> **A2:** Thank you for raising this important concern. We believe there may be some misunderstanding about our workflow, so please allow us to clarify the operational phases and scalability characteristics of DARGO:
>
> **On retrieval latency:**
> Our framework operates in two distinct phases: **exploration** and **inference**. During exploration, we construct a memory database by storing successful cases generated through LLM-SLM orchestration on the training set. Crucially, once exploration is complete, this database becomes **frozen** and remains static throughout all following inference operations for unseen tasks in testing. During inference, each unseen testing tasks follows an identical workflow: retrieve exactly 3 case studies from the fixed database → synthesize meta-instructions → generate code. This design ensures that inference latency is **orthogonal** to task volume growth because the database size, retrieval count, and processing workflow remain constant regardless of the number of evaluation testing examples processed. It will be the case that retrieval latency increases as the number of cases in the database increases as there are more entries to consider, but this is true of any RAG-based system that employs a vector database. We do not expect this to be a bottleneck due to its efficient indexing, especially when compared to SLM inference latency of any system.
>
> **On storage demands:**
> 1) As discussed in Section 1, high-quality training data for DACG is inherently scarce. Even in our largest experimental setting (1,000 training samples), our exploration phase retains only successful cases, resulting in compact memory databases. Despite this low-resource constraint, our experiments demonstrate encouraging performance improvements.
> 2) For adopting our method with larger datasets, scalability can be easily managed by implementing a maximum memory database size limit, ensuring consistent retrieval latency regardless of training set scale.
>
> Notably, we maintain strict experimental integrity by **NEVER** incorporating test cases into the memory database during inference, preventing any unfair advantage from ground truth leakage while ensuring fair comparative evaluation. Therefore, regardless of the volume of inference tasks processed, latency and storage concerns remain constant and do not pose scalability bottlenecks.
>
> > Concern 3: Focuses exclusively on Python/SQL for tabular data tasks, while other languages or code-generation scenarios remain unexplored.
>
> **A3:** Thanks for your suggestion. We provide our response here:
>
> 1) DACG represents a complex and representative domain that requires both code generation capability and data understanding capability, making it a rich testbed for knowledge distillation beyond syntax. Success in DACG demonstrates the ability of our work to handle tasks requiring multi-step reasoning and domain-specific expertise.
>
> 2) DACG has a significant real-world impact and ubiquity. Data analysis and code generation are crucial needs across industries and research. Improving DACG directly benefits data scientists, researchers, and analysts in their daily work especially since the data covers a wide range of domains such as analysis in biomedical science, sports, finance, education.
>
> 3) DACG introduces unique challenges are not as prevalent in other code generation domains. For example, a key challenge that we found and addressed was handling heterogeneous data contexts and schema during distillation. If the distillation was too specific to the schema, it would not generalize, confusing the SLM during inference. Existing work that distils helper functions (e.g. ReGAL) created functions that were coupled too specific to domains and thus not generalizable. By focusing on DACG we are able to explore the challenges in depth.
>
> Further, we summarized from open survey to prove research about Python and SQL can cover most usage in the real world:
> 1. Stack Overflow's 2024 Developer Survey (65,000+ respondents) shows that 73% of data professionals use Python regularly and 60% regularly use SQL.
> 2. Industry reports indicate that over 85% of data job postings mention SQL as a must-have skill 70% of data roles demand SQL knowledge according to DataCamp surveys.
>
> > Concern 4: Relying on closed-source LLMs as teachers still remain privacy and access concerns.
>
> **A4:** Thanks for asking. However, we think there are some misunderstanding. We believe this design choice actually strengthens rather than conflicts with our privacy motivation.
>
> 1)  The key insight is the separation between exploration and inference phases. During the exploration including MOI stage, memory construction, and case study translation, teacher LLMs work with "training" data to build the memory database (similar to how university students attend classes with teachers for learning). However, during inference for new queries in testing, only the SLM operates locally using the pre-constructed memory database, like students taking exams independently using their learned knowledge. This ensures that sensitive user data and queries **NEVER** interact with external models in testing, maintaining complete privacy during deployment. Once the memory database is constructed, the entire system can operate offline in privacy-sensitive environments without any external dependencies.
>
> 2) Additionally, we show the performance of DarGO when the teacher LLM is replaced with an open-weight and popular model **Llama 3.3 70B**. Section 3.4 (paragraph) with the evidence in Table. 3, highlighting that DarGO remains effective with purely open models. This supports our claim that we are proposing an effective knowledge distillation workflow, instead of relying on specific proprietary models on both teacher and student ends.
>
> > Question 1: How do you define small language model and large language model? How to distinguish them?
>
> **A5:** Thank you for this important clarification question.
> Here are our operational definition of Small Language Models (SLMs):
> We define SLMs based on two key criteria mentioned in Lines 207-210:
>
> 1) Computational constraint: Models with ≤15B parameters that can be efficiently deployed on a single high-end GPU (A100/A800). This threshold reflects practical deployment considerations for resource-constrained environments.
> 2) Emergent capability (ICL) threshold: Models must demonstrate in-context learning (ICL) capabilities, including instruction following and few-shot adaptation without requiring supervised fine-tuning for new tasks. In such context, models like T5 or BERT, despite having fewer parameters, lack the instruction-following and ICL capabilities we associate with modern language models, and thus fall outside our SLM definition.
>
> > Question 2: Which BIRD setting are you using? Is Oracle Knowledge provided to the model? This is a huge difference.
>
> **A2:** Thanks for asking this. Yes we use Oracle Knowledge by appending it to question as part of user task query which is a common or default usage for this dataset. Observing BIRD-SQL leaderboard, almost all works include Oracle Knowledge into input. But we will add this detail in revision if we get a chance. Thanks for highlighting this and improving the rigor of the paper. Thank you!

---

> > ### Comment · Reviewer_Y9rz · 2025-08-05
> > **Response**
> >
> > Why do you say `This design ensures that inference latency is orthogonal to task volume growth`? Are you saying you achieve O(1) time complexity for a retrieval algorithm?

---

> ### Author Response · Authors · 2025-08-05
> **Thanks for the Discussion!**
>
> Dear Reviewer,
>
> We thank you for your engagement and for posing this important clarification question. We are **not** claiming O(1) time complexity for our retrieval algorithm. Let us clarify what we mean by "inference latency is orthogonal to task volume growth." according to our understanding.
>
> We guess there maybe a potential misunderstanding of the term **"task volume growth"** in the context of our work.
>
> * ***Our Definition of "Task Volume Growth"***: We use this term to refer to the number of inference queries for testing that need to be processed (e.g., evaluating on 100 vs. 1,000 test queries).
>
> * ***Our Meaning of "Orthogonal"***: We state that per-query inference latency is "orthogonal" in this context because the processing time for each individual query remains constant, regardless of the total number of queries being evaluated. This is due to several factors:
>     1.  **Fixed Memory Database Size**: After the initial exploration phase, our memory database is frozen. Its size is fixed and does not increase as more test queries are processed.
>     2.  **Constant Retrieval Operations**: For every inference query, the system retrieves a fixed number of case studies (`k=3`) from this static database.
>     3.  **Identical Workflow**: Every query follows the same `retrieve → synthesize Meta-Instruction → generate code` pattern, ensuring the per-query processing time is consistent. In this sense, per-query latency does not grow with the volume of test queries.
>
> Furthermore, we wish to clarify that while Retrieval-Augmented Generation (RAG) is a foundational component of our pipeline, our core novelty lies in the proposed tuning-free distillation method for complex DACG tasks. As shown in our submission code (`src/rag_copy.py`), our RAG implementation is based on a standard FAISS library with its default k-NN algorithm.
>
> Given this implementation, the computational complexity is as follows:
>
> * There is a one-time, linear cost of **O(N)** to build the vector index for the N items in the searchable knowledge base.
> * After indexing, each of the M queries is processed with a highly efficient retrieval time of approximately **O(logN)**, due to the FAISS search index.
>
> The practical impact of this retrieval phase on overall latency is minimal. For example,  for the TabMWP dataset, building the vector index required **7.59** seconds for 866 successful cases (which is fixed, one-time), while the retrieval phase (also including appending 3 neighbors and write into data) took **11.57** seconds for 1000 test cases.
>
> Further, as shown in Table 1 about results and statistics in Table 6, that our database has < 1000 entries, but Whilst we might want to make this larger, we have demonstrated strong performance improvements at this scale. We think it already proves our motivation that there is a way to do training-free knowledge distillation for DACG. And we think how to further optimize retrieving phase would be future work of RAG domains.

---

> > ### Comment · Reviewer_Y9rz · 2025-08-05
> > **Response**
> >
> > Thank you for your reply.
> > I think your response is somewhat evasive. For a serial algorithm, the time for each sample is nearly the same for both 100 samples and 1000 samples isn't anything new to mention; it's just common sense.
> > I understand your claim now and I'll keep my score, thanks for your clarification.

---

> > > ### Author Response · Authors · 2025-08-05
> > >
> > > Thank you for your quick reply. We did not intend to be evasive. Our goal was to make sure we understood the stated weakness correctly and your interpretation of 'task volume growth'. If we have misinterpreted we appreciate the opportunity to clarify. We agree that time per sample in a serial algorithm is usually invariant, but we are otherwise unsure how to interpret the terminology.
> > >
> > > As we include in response, we acknowledge that retrieval increases with database size, but in a manner that is standard for any RAG system. Our claim is that this is no more of a bottleneck than any typical RAG system.
> > >
> > > We hope that our earlier response has addressed the issues you identified, namely:
> > > - W1. Cost: A comprehensive cost analysis with comparable approach was included in the submission.
> > > - W2. Discussed above.
> > > - W3. We acknowledge that the presentation of our work is limited to this domain, but highlight that the significance of this domain in real-word applications means it is worth exploring independently, as evidenced by the significant existing literature that focuses on data analysis / python code generation.
> > > - W4. Our system does not rely on closed sourced models as we also evaluate using llama as a teacher model, included in Table3.
> > >
> > > We would appreciate any further discussion as to whether these points adequately address your concerns. Thank you again for your review and response.

---

### Official Review · Reviewer_nVTU · 2025-07-03

**Clarity:** 3
**Significance:** 2
**Originality:** 3
**Rating:** 4
**Confidence:** 4

**Summary:**

The paper introduces a training free knowledge distillation approach to make SLMs better at data analysis code generation tasks. They achieve this through the curation of synthetically generated functional plans from a teacher model that are adapted to the student. These plans are used as context during inference with the help of RAG. It evaluates this approach on multiple publicly available datasets and models.

**Questions:**

- In the results shown in Table 9, is the LoRA model fine-tuned on the original data, or on the constructed database that is used during RAG?
- The result tables in the main paper are missing standard errors. Is this data included elsewhere?

**Ethical Concerns:**

["NO or VERY MINOR ethics concerns only"]

**Final Justification:**

Author rebuttal addressed most of my concerns. The overall pipeline shows benefits with ablations to justify each component. My criticisms for this paper:
1. Evaluating on more varied domains will strengthen the paper if successful. Nothing in the pipeline is very specific to DACG.
2. As mentioned by another review, I agree that the method is a bit engineering heavy.

**Limitations:**

yes

**Paper Formatting Concerns:**

None.

**Quality:**

3

**Strengths And Weaknesses:**

Strengths:
- The approach to use the synthetically generated data in-context without fine-tuning is quite useful and avoids expensive training runs.
- MOI (Section 2.2) and the ablation experiment from Table 5 show the effectiveness of SLM participation during the planning phase.
- The paper also shows the generalization of the generated knowledge by using it with other student models (Section 3.4).

Weaknesses:
- The experiments mainly focus on non-reasoning SLMs like Phi-3, but omit recent reasoning specialized SLMs. Gain in results for such models might differ, since they are already trained for task decomposition in general purpose scenarios.
- Since the paper focuses on a training-free approach, it would be useful to see a proper comparison between fine-tuning and RAG, by using the same constructed memory database for fine-tuning.
- Evaluation scope is limited to data analysis tasks. This approach is general enough that it could work with a broader set of tasks.
- The discussion on the broader impact and future work would be more useful if moved to the main paper.

---

> ### Author Rebuttal · Authors · 2025-07-30
>
> > Concern 1: The experiments mainly focus on non-reasoning SLMs like Phi-3, but omit recent reasoning specialized SLMs. Gain in results for such models might differ, since they are already trained for task decomposition in general purpose scenarios.
>
> **A1:** Thanks for raising this. The release of many of these models, such as Qwen-3 with reasoning capability, coincided with submission of this work. (Qwen-3 released 14th May, and Phi-4-Reasoning appeared on 30th April). We do agree that reasoning SLMs may exhibit different interactions and are important to consider. As a result, we conduct additional experiments of all settings for qwen-3-8B with reasoning enabled as:
>
> | Model (Qwen3-8B) | BIRD-SQL | WIKITQ | TabMWP | CRTQA | QrData | INFI-AGENT |
> |------------------|----------|--------|--------|-------|--------|------------|
> | Zero-shot        | 41       | 37.5   | 45.1   | 40.52 | 44.05  | 47.13      |
> | + DarGO           | 47.2     | 42.8   | 48.8   | 46.7  | 48.86  | 53.7       |
>
> As we can observe, our method can be successfully applied to recent reasoning based SLMs with no additional modification of the pipeline. The performance of Qwen3-8B with reasoning enabled powered by DarGO is the best on BIRD-SQL (Table 1) with 47.2%. We thank for your suggestions and curiosity, we will add them into revision if possible.
>
> > Concern 2: In the results shown in Table 9, is the LoRA model fine-tuned on the original data, or on the constructed database that is used during RAG? Since the paper focuses on a training-free approach, it would be useful to see a proper comparison between fine-tuning and RAG, by using the same constructed memory database for fine-tuning.
>
> **A2:** Thanks for asking this. We already have comparison with RAG based approach in which dynamic few shot in Table 1 was implemented by retrieving relevant examples from our output memory database as few shot examples for augmentation of generation. The FT results in Table 9 are based on same initial training samples, not the samples from the memory database since we considered **auto-generated memory database** as our contribution. However, we agree that SFT on the memory database would strengthen our work. We present experiment and list result here:
> | Model | SIMPLE | MEDIUM | CHALLENGING | OVERALL |
> |-------|--------|--------|-------------|---------|
> | | ***Zero-Shot End-to-End Code Gen.*** | | | |
> | Original Checkpoint | 38.51 | 21.20 | 11.76 | 24.40 |
> | LoRA Fine-Tuned | 39.86 | 19.20 | 10.78 | 23.60 |
> | SFT MEM | 40.12 | 22.80 | 12.45 | 27.20 |
> | | ***DARGO Knowledge Distillation*** | | | |
> | Meta Instruction | 51.35 | 30.40 | 16.67 | 33.80 |
>
> These results demonstrate that our memory database provides substantial benefits even for traditional fine-tuning approaches. We attribute this improvement to two key factors:
>   1. our Model Orchestration Interface (MOI) effectively filters noisy or low-quality data, including ground truth errors that may mislead training
>   2. the orchestrated codes in our memory database contain structured comments and planning information that enhance reasoning capabilities, contrasting with original declarative ground truth codes that may impose additional inferential burden on SLMs during learning.
>
> As we illustrate in D.2, we hypothesize that fine-tuning high quality data may exceed ICL when the samples are large enough.
>
> Notably, while SFT on our memory database shows improvement over baseline fine-tuning, our training-free knowledge distillation approach still achieves superior performance while avoiding the computational overhead of parameter updates.
>
> > Concern 3: Evaluation scope is limited to data analysis tasks. This approach is general enough that it could work with a broader set of tasks.
>
> **A3:** Thanks for your suggestion and acknowledgement of the generality of this approach. We agree that it can be applied to a broader set of tasks, however we focus on data analysis code generation (DACG) due to its significance and challenge. By focusing on DACG we can provide more in-depth analysis and comparison with other approaches. Specifically:
>
> 1) DACG represents a complex and representative domain that requires both code generation capability and data understanding capability, making it a rich testbed for knowledge distillation beyond syntax. Success in DACG demonstrates the ability of our work to handle tasks requiring multi-step reasoning and domain-specific expertise.
>
> 2) DACG has a significant real-world impact and ubiquity. Data analysis and code generation are crucial needs across industries and research. Improving DACG directly benefits data scientists, researchers, and analysts in their daily work especially since the data covers a wide range of domains such as analysis in biomedical science, sports, finance, education.
>
> 3) DACG introduces unique challenges are not as prevalent in other code generation domains. For example, a key challenge that we found and addressed was handling heterogeneous data contexts and schema during distillation. If the distillation was too specific to the schema, it would not generalize, confusing the SLM during inference. Existing work that distils helper functions (e.g. ReGAL) created functions that were coupled too specific to domains and thus not generalizable. By focusing on DACG we are able to explore the challenges in depth.
>
> > Concern 4: The discussion on the broader impact and future work would be more useful if moved to the main paper.
>
> **A4:** Thanks for your suggestion, we will do this in revision.
>
> > Concern 5: The result tables in the main paper are missing standard errors. Is this data included elsewhere?
>
> **A5:** Thanks for asking this. For each result, we conduct experiment 5 times and report averaged results for neat presentation in tables as we mentioned in Checklist Section 7 (L. 730-732) with temperature as 0.0 and set top_p as 1 so the performance is stable. But we also summarize this statistics for methods across datasets here:
>
> | Dataset | Baseline Methods | DARGO Methods |
> |---------|------------------|---------------|
> | **BIRD-SQL** | ±0.3 | ±0.1 |
> | **WikiTQ** | ±1.5 | ±1.1 |
> | **TabMWP** | ±1.3 | ±0.9 |
> | **CRTQA** | ±1.2 | ±0.8 |
> | **QrData** | ±0.1 | ±0.2 |
> | **INFI-AGENT** | ±0.8 | ±0.3 |

---

> ### Comment · Area_Chair_T6r6 · 2025-08-06
> **Discussion Period Ending Soon**
>
> Dear Reviewer,
>
> The discussion period is ending soon. We would be grateful if you could take a moment to review the authors' response to your comments and provide any final feedback.
>
> We truly appreciate your time, effort, and valuable contributions to the review process.
>
> Best regards,
>
> AC

---

> ### Comment · Reviewer_nVTU · 2025-08-06
>
> I thank the authors for their additional results and clarifications. These have addressed most of my concerns and I have updated my score.

---

> > ### Author Response · Authors · 2025-08-07
> > **Thanks for your time!**
> >
> > Thank you for the discussion. We are pleased our response addressed your concerns and appreciate your time and feedback.

---

### Official Review · Reviewer_Nr8J · 2025-07-03

**Clarity:** 1
**Significance:** 2
**Originality:** 2
**Rating:** 4
**Confidence:** 4

**Summary:**

This paper introduces DARGO, a training-free framework for distilling knowledge from Large Language Models (LLMs) to Small Language Models (SLMs) in the context of Data Analysis Code Generation (DACG) tasks. Instead of relying on resource-intensive fine-tuning, DARGO uses In-Context Learning (ICL) to transfer knowledge through a three-phase process: model orchestration, memory collection, and knowledge-guided inference. Experiments on benchmarks such as WIKITQ, TABMWP, and BIRD-SQL show that DARGO significantly improves SLM performance, with an average relative gain of 27.5%. The paper also explores generalization across different model pairings and transfer scenarios, offering a novel and efficient approach to LLM-to-SLM distillation.

**Questions:**

N/A

**Ethical Concerns:**

["NO or VERY MINOR ethics concerns only"]

**Final Justification:**

The author's rebuttal has addressed my major concerns, so I decided to increase my rating from 2 to 4.

**Quality:**

2

**Strengths And Weaknesses:**

Strengths
1. The paper tackles an important and timely problem—efficiently distilling large models into small ones using a training-free approach, which is highly practical.
2. The framework design is clear and well-structured, with a logical three-phase pipeline.
3. Empirical results are strong, showing consistent improvements across multiple benchmarks and model architectures.

Weaknesses
1. The method is engineering-heavy, relying on many handcrafted components, which limits theoretical insight and generalizability.
2. Comparative experiments are insufficient—there is no evaluation against strong recent code generation models such as Qwen2.5-3B or other modern coder LLMs (at the same setting). Besides, Table 1 only shows the improvement of the proposed method, lacking comparison with other methods that also focus on knowledge distillation.
3. The presentation quality is poor—several figures are unclear or hard to read, which affects the overall readability of the paper.

---

> ### Author Rebuttal · Authors · 2025-07-30
>
> > Concern 1: The method is engineering-heavy, relying on many handcrafted components, which limits theoretical insight and generalizability.
>
> **A1:**  Our approach does involve rigorous engineering, however we believe this is essential for real-word challenges in data analysis, and the **application** track under which we have submitted. Our system is intentionally designed, "with thorough ablation studies validating each module." (Reviewer Y9rz). Specifically, we may not agree that this limits the insight and **generalizability** of our work.
>
> To highlight some key insights from our work:
>
> **MOI, or teacher-student planning**
>
> Through ablation we show that generated executable plans through iteration between teacher and student is superior to teacher generated plans (Table 5). This is because of a mismatch in granularity between teacher generated plans, and the level of detail required by the student model.  Our work indicates that the student should be involved in the modularization process to correctly align the granularity of the transformation. This is a key insight that we believe is novel and not present in prior work.
>
> **Semantics over syntax**
>
> A further key insight from our works supported via ablation (Table 5) is the role of case studies, or summarizing the demonstrations in a textual and semantic way, rather than being schema, syntax, or example driven. This is especially crucial in the data analysis context where the data is **heterogeneous** and the schema the schema may vary between each task. When compared with other approaches that are syntax based, such as ReGAL (ICML 2024) that generates reusable helper functions, we find that our approach performs better. We believe the concept of case-study summarization is novel in our work, and is rigorously validated across multiple different benchmarks and languages.
>
> **On generalization**
>
> For generalization, we conduct comprehensive experiments in Section 3.4 demonstrating that our framework successfully transfers across different programming languages (Python, SQL), data modalities (tabular, relational databases), and teacher (GPT-4o, Llama 3.3 70B) /  student models (Llama-3.1-8B, Phi-3-mini, starcoder), as evidenced in our cross-model experiments (Tables 2-3) and both in-domain and out-of-domain evaluations. As part of our response we also show that DARGO generalizes to SLMs with reasoning capabilities. Aside from minor modifications to support different output languages and execution, our system is unchanged across all these configurations, therefore we do not believe that our method inhibits generalizability.
>
> Additionally, we contend that substantial engineering effort is unavoidable and necessary in the application track. Consider established baselines that have been accepted at top-tier venues: DSPy (ICLR) and ReGAL (ICML), both of which inherently contain handcrafted components. Compared to these methods, our approach actually requires **less** engineering overhead while being more **cost-effective**, as demonstrated by our comprehensive cost analysis across all procedures in Appendix I, where DARGO proves to be the most resource-efficient solution.
>
> > Concern 2: Comparative experiments are insufficient—there is no evaluation against strong recent code generation models such as Qwen2.5-3B or other modern coder LLMs (at the same setting).
>
> **A2:** We appreciate the suggestion to include more specific models, including code generating and reasoning models. The nature of the field is such that new models will always be released post-submission. Our primary contribution is demonstrating effective training-free knowledge distillation, as acknowledged by Reviewer xWYB, rather than showcasing specific model performance.
>
> We show consistent improvements across diverse architectures that were current at the time of submission, validating our approach's robustness. Concretely, for teacher models, we conduct the same auto pipeline for GPT-4o and Llama 3.3 70B; for student models, we test Llama 3 8B, phi-3-mini, starcoder 7B. Additionally, we provide extensive ablation studies (Section 3.5) and out-of-distribution evaluations (Section 3.4) that demonstrate the method's effectiveness beyond simple performance comparisons.
>
> However, we appreciate your suggestions for experiments for very recent SLMs, and acknowledge that reasoning models are interesting to study. Here we evaluate DarGO on Qwen 2.5 Coder in 1.5B and 7B, and a very recent reasoning model Qwen-3-8B with reasoning enabled. Results:
>
> | Model (Qwen2.5-Coder-1.5B) | BIRD-SQL | WIKITQ | TabMWP | CRTQA | QrData | INFI-AGENT |
> |---------------------------|----------|--------|--------|-------|--------|------------|
> | Baseline                 | 22.4     | 28.5   | 35.8   | 25.27 | 43.29  | 42.8       |
> | + DarGO                     | 30.2     | 35.2   | 48.5   | 42.86 | 48.61  | 46.3       |
>
> | Model (Qwen2.5-Coder-7B) | BIRD-SQL | WIKITQ | TabMWP | CRTQA | QrData | INFI-AGENT |
> |--------------------------|----------|--------|--------|-------|--------|------------|
> | Baseline               | 37.2     | 36.8   | 41.2   | 34.75 | 48.86  | 51.36      |
> | + DarGO                    | 43.4     | 44.1   | 52.8   | 43.82 | 50.13  | 54.47      |
>
> | Model (Qwen3-8B) | Mini-Dev | WIKITQ | TabMWP | CRTQA | QrData | INFI-AGENT |
> |------------------|----------|--------|--------|-------|--------|------------|
> |Baseline        | 41       | 37.5   | 45.1   | 40.52 | 44.05  | 47.13      |
> | +DarGO            | 47.2     | 42.8   | 48.8   | 46.7  | 48.86  | 53.7       |
>
> The consistent performance improvements across these recent models demonstrate that DARGO generalizes effectively, including specialized code generation models and reasoning-enhanced variants, further validating the statement of the paper.
>
> > Concern 3: Besides, Table 1 only shows the improvement of the proposed method, lacking comparison with other methods that also focus on knowledge distillation.
>
> **A3:** We do think there are some misunderstandings for this concern. **In the 4th and 5th line of SLM performance in Table 1**, we compare our performance with one popular distillation pipeline **DSPy** (ICLR 2024) for general NLP tasks, a function-based distillation pipeline for code generation **ReGAL** (ICML 2024).
>
> Section 3.3 (2) gives a comparison with these approaches and we further present a cost analysis in Appendix I. Our results show that DARGO is more effective and cost efficient. A key insight from our analysis is that DARGO is more robust due to its distillation approach. DSPy is few-shot based that can overfit or be effected by noisy data. ReGAL generates reusable python helper functions, however these are tightly coupled to specific domains and schema (e.g. `filter_senator_by_year`). Our contribution of case-study summarization and instruction based distillation mitigates these issues.
>
> > Concern 4: The presentation quality is poor—several figures are unclear or hard to read, which affects the overall readability of the paper.
>
> **A4:** Thanks for raising this. We appreciate that some figures could benefit from enhancement and clearer graphics, and will address this in revision.

---

> > ### Comment · Reviewer_Nr8J · 2025-08-05
> >
> > I appreciate the authors' detailed response, which partially resolved my concerns. Regarding the experiment part, there are two further questions arising: (1) What is the detailed training data and evaluation setting of the newly introduced experiments with Qwen2.5-Coder and Qwen3? (2) How about taking Qwen2.5-Coder as baseline methods and comparing with Qwen2.5+DRAGO?

---

> ### Author Response · Authors · 2025-08-05
> **Thanks for your Reply!**
>
> Dear Reviewer,
>
> Thank you for your continued engagement with our work and for reading our rebuttal. We appreciate the opportunity to clarify these points.
>
> Here are our answers to your follow-up questions:
>
> > **(1) What is the detailed training data and evaluation setting of the newly introduced experiments with Qwen2.5-Coder and Qwen3?**
>
> **A5:** Thank you for asking for these details. The experimental setup for the new Qwen models is identical to the main experiments described in our paper. The seed data and evaluation settings remain the same - we **only** change inference endpoint for the student model and run our pipeline again. To be more precise:
>
> * **No Parameter Updates:** We use the same seed data to initialize the process for DarGO. No model parameters are updated during this process; the orchestration and distillation are performed entirely through In-Context Learning.
> * **Default Setup:** We followed the setup detailed in Section 3.2 and Appendix B. We simply replaced the student model with the new Qwen2.5-Coder and Qwen3 models for the results provided in the rebuttal.
> * **Teacher Model:** The teacher model remains GPT-4o due to its strong performance and accessibility. Our ablation in Table 3 (using Llama-3.3 70B as a teacher) demonstrates that DarGO's effectiveness is not dependent on a specific proprietary teacher model.
>
> > **(2) How about taking Qwen2.5-Coder as baseline methods and comparing with Qwen2.5+DRAGO?**
>
> **A6:** Thank you for this question;  Our table shows comparison `Qwen2.5-Coder` and `Qwen2.5-Coder + DarGO` (with performance increases across multiple model sizes). We did not compare `Qwen2.5-Coder` and `Qwen2.5 instruct+ DarGO` because the experiment has two variables (model/pre-training variant and DarGO) and it would be difficult to draw insight from. In trying to help us align with NeurIPS guidelines where discussion is intended for clarification of the existing work, rather than new experiments,  could we ask the following questions to understand how to clarify the presentation of our results:
> - What specific aspect of our effectiveness of our method does our current experimental evidence not adequately demonstrate?
> - Which claims or conclusions in our paper would benefit from this additional comparison to strengthen the evaluation?
>
> We would add this experiment if this could change your mind and can resolve misunderstanding in our paper.
>
> We do agree that showing that DARGO works for multiple model variants is important: coder, instruct, reasoning. Our results present this across multiple models, including recent ones such as Qwen2.5 coder, Qwen3 reasoning, Phi3, Llama-3.1-8B. Here the only variation is DARGO which increases performance in each case.
>
> Thanks again for your time in reading our paper and rebuttal!

---

> ### Author Response · Authors · 2025-08-07
> **Follow-up Response**
>
> Dear Reviewer,
>
> Thanks for your time in reviewing our work and giving comments. Here are results comparing Qwen Coder as baselines and Instruct models + DarGO to further validate our statement as you suggested:
>
> ### For 1.5B Models:
> | Model | BIRD-SQL | WIKITQ | TabMWP | CRTQA | QrData | INFI-AGENT |
> |-------|----------|--------|--------|-------|--------|------------|
> | Qwen2.5-Coder-1.5B | 22.4 | 28.5 | 35.8 | 25.27 | 43.29 | 42.8 |
> | **Coder + DARGO (Ours)** | **30.2** | **35.2** | **48.5** | **42.86** | **48.61** | **46.3** |
> | Qwen2.5-1.5B-Instruct | 16.8 | 16.2 | 30.1 | 16.21 | 30.13 | 33.07 |
> | **Instruct + DARGO (Ours)** | **28.2** | **30.8** | **42.8** | **30.77** | **32.40** | **42.8** |
>
> ### For 7B Models:
> | Model | BIRD-SQL | WIKITQ | TabMWP | CRTQA | QrData | INFI-AGENT |
> |-------|----------|--------|--------|-------|--------|------------|
> | Qwen2.5-Coder-7B | 37.2 | 36.8 | 41.2 | 34.75 | 48.86 | 51.36 |
> | **Coder + DARGO (Ours)** | **43.4** | **44.1** | **52.8** | **43.82** | **50.13** | **54.47** |
> | Qwen2.5-7B-Instruct | 32.0 | 32.3 | 39.4 | 32.28 | 37.72 | 49.42 |
> | **Instruct + DARGO (Ours)** | **40.2** | **47.1** | **47.5** | **47.12** | **46.84** | **54.09** |
>
> We can observe that DarGO can also optimize performance for non-coder instruct model and can even achieve better performance than Coder as baselines. Rather than requiring specialized training (as in Coder variants), DarGO achieves comparable performance through training-free knowledge distillation, making it more accessible and cost-effective for DA researchers.
>
> The consistent improvements across both Instruct and Coder variants (as shown in our original results) further validate our robustness and generalizability of our approach.
>
> Thanks again for your engagement of discussion!
>
>
> Best, \
> Authors

---

### Official Review · Reviewer_xWYB · 2025-07-03

**Clarity:** 3
**Significance:** 3
**Originality:** 3
**Rating:** 5
**Confidence:** 3

**Summary:**

This paper proposes DARGO, a training-free knowledge distillation framework that enhances small language models (SMLs) for data analysis code generation. DARGO is composed of three main components: 1) it constructs solution examples through Model Orchestration Interface (MOI) to orchestrate LLM-SLM collaboration, 2) it filters out correct solutions and condenses them into general reasoning patterns stored in the Memory Database, and 3) it retrieves relevant examples from the Memory Database using RAG for In-Context Learning demonstrations for SMLs. The authors conduct extensive experiments to demonstrate the effectiveness of DARGO.

**Questions:**

1. The retriever model (i.e., the embedding model) is not perfect. If the retrieved examples are not relevant to the query, could it mislead the SLM and cause a performance drop?
2. Since the DACG performance of the SML is heavily dependent on the quality and diversity of the Memory Database, does this mean that a large number of examples need to be constructed during the MOI stage? One of the motivations for this paper is to address privacy concerns when using closed-source models. Would requiring large-scale LLM calls during the MOI stage conflict with this motivation?
3. In the error analysis, 54% of errors are attributed to "over-reasoning." Are there any proposed solutions to this?

**Ethical Concerns:**

["NO or VERY MINOR ethics concerns only"]

**Limitations:**

yes

**Quality:**

3

**Strengths And Weaknesses:**

Strengths:
1. The writing is clear, and the motivation is well-defined.
2. The experiments are thorough, including a wide range of analytical experiments that convincingly prove the effectiveness of DARGO.
3. The authors have submitted the complete code.

Weaknesses:
See "Questions" below.

---

> ### Author Rebuttal · Authors · 2025-07-30
>
> > Q1: The retriever model (i.e., the embedding model) is not perfect. If the retrieved examples are not relevant to the query, could it mislead the SLM and cause a performance drop?
>
> **A1:** Thank you for this question. We think DARGO is specifically designed to be robust against retrieval noise through several key mechanisms:
> 1. DARGO doesn't use raw retrieved cases directly. Instead, it transforms them through **Case Study** translation (Section 2.3), which extracts generalizable reasoning patterns and illustrate how to correctly plan for solving DA tasks rather than specific implementations, making the system inherently robust to individual case quality.
> 2. Our meta-instruction (MI) generation process aggregates knowledge across multiple retrieved cases, effectively filtering out noise through consensus-based distillation (Section 2.4).
> 3. Our results demonstrate consistent improvements across all datasets and model combinations, indicating that the framework successfully handles retrieval imperfections in practice. Table 10 (Appendix E) presents an ablation focused on embedding and retrieval. Including concrete code examples can bias the SLM (Instruction Type w/ examples), but we mitigate this in DARGO by the distillation of more general and transferrable instructions and case studies.
> 4. DARGO introduces novel techniques that mitigate this, but we agree it is not a solved problem. Further promising approaches to reducing such noise would be a RAG implementation by Top K + Top P, which means after retrieval, we should set a value of p as a threshold for filtering those irrelevant.
>
>
> > Q2: Since the DACG performance of the SML is heavily dependent on the quality and diversity of the Memory Database, does this mean that a large number of examples need to be constructed during the MOI stage?
>
> **A2:** Thanks for asking this. We argue that DARGO is a data-efficient knowledge distillation pipeline, without data expansion such techniques as self-evolve, based on the following:
> 1. All experiments use modest, fixed datasets (1,000 examples each), demonstrating that DARGO maximizes value from limited data rather than requiring large-scale construction.
> 2. The MOI stage refines existing examples and appends SLM-friendly reasoning steps through orchestration and optimization, improving quality rather than increasing quantity.
> 3.  As shown in Appendix D.2, DARGO significantly outperforms traditional lora fine-tuning approaches while using the same limited training data, demonstrating its data efficiency. This efficiency is precisely what makes DARGO valuable for DACG domains where high-quality annotated data is inherently scarce. Also, as shown in Table 12, 14, 16, we show that DARGO incurs less costs compared to popular distillation pipelines such as DSPy and ReGAL, while achieving better performance.
>
> > Q3: One of the motivations for this paper is to address privacy concerns when using closed-source models. Would requiring large-scale LLM calls during the MOI stage conflict with this motivation?
>
> **A3:** Thank you for raising this consideration. We believe this design choice strengthens rather than conflicts with our privacy motivation. The key insight is the **separation** between exploration and inference phases. During the exploration phase that includes MOI, memory construction, and case study translation, teacher LLMs work with "training" data to build the memory database (similar to how university students attend classes with teachers for learning). These training samples could be acquired from open domains or manually curated in the real-world implementation. In this work, we used small portions of official training split in each benchmark as other previous works did. However, during inference for new queries at test-time, only the SLM operates locally using the pre-constructed memory database, like students taking exams independently using their learned knowledge. This ensures that sensitive user data and queries never interact with external models in testing, maintaining complete privacy during deployment. Once the memory database is constructed, the entire system can operate offline in privacy-sensitive environments without any external dependencies, and this memory database is like a treasure which can be used in multiple downstream tasks. And we prove its generalization in the experiment parts. Further, we also show that DARGO also works with open teacher models (Table 3).
> Thanks for asking this.
>
> > Q4: In the error analysis, 54% of errors are attributed to "over-reasoning." Are there any proposed solutions to this?
>
> **A4:** Thank you for this insightful observation and does highlight an opportunity for future work and improvements. In Appendix G, we envision several promising directions to address this. A task complexity router could dynamically determine when reasoning is beneficial versus when direct generation suffices. This router could be implemented as a lightweight classifier or leverage the SLM's own assessment capabilities based on memory database patterns. Additionally, enriching the memory database with diverse examples across different complexity levels may also help. These extensions represent natural evolution paths that could further optimize the reasoning-efficiency trade-off while preserving DARGO's core advantages. We would consider this as a promising direction in the future. Thanks!

---

> > ### Comment · Reviewer_xWYB · 2025-08-07
> >
> > Thanks for the detailed explanation. I do not have further questions.

---

> ### Comment · Area_Chair_T6r6 · 2025-08-06
> **Discussion Period Ending Soon**
>
> Dear Reviewer,
>
> The discussion period is ending soon. We would be grateful if you could take a moment to review the authors' response to your comments and provide any final feedback.
>
> We truly appreciate your time, effort, and valuable contributions to the review process.
>
> Best regards,
>
> AC

---

> ### Author Response · Authors · 2025-08-07
> **Thanks for your Feedback**
>
> Dear Reviewer,
>
> Thanks for your comments and we appreciate you taking the time to provide feedback!
>
> Best, \
> Authors

---

### Note · Authors · 2025-08-11

Based on our submission and the valuable feedback and discussion with reviewers, we summarize our contributions as a **cost-effective** distillation approach that works for **multiple models** including recent **coding and reasoning SLMs**, and **does not rely on closed-source teacher models**. Specifically, reviewers highlighted the following concerns that we have comprehensively addressed:

**Model generalization:**

We show that DARGO generalizes effectively to recent models including Qwen2.5-Coder (1.5B and 7B) and Qwen3-8B with reasoning capabilities. Our results demonstrate consistent improvements in new suggested models.

**Independence from closed-source models:**

Our approach is not reliant on closed-source teacher models. We show that DARGO works effectively when using the open-source Llama-3.3-70B model as a teacher (Table 3), achieving comparable results to GPT-4o. A further concern that was raised around privacy, which we address in two ways:
1. Using open-source teacher models.
2. Our separation of exploration and inference phases ensures that sensitive user data never interacts with external models during deployment. Only public or non-sensitive data is used during the exploration phase that builds the memory database.

**Cost Effectiveness & Efficiency:**

Our comprehensive cost analysis demonstrates that DARGO is the most resource-efficient among distillation methods we evaluated, requiring significantly lower costs than DSPy and ReGAL (Appendix I) while achieving better performance. We also show a detailed RAG latency analysis in note XAKuEnoFxT (first response to Reviewer Y9rz)

**Applicability and generalization:**

Our work demonstrates broad applicability across main DA programming languages (Python, SQL), data modalities (tabular, RDBs), and model combinations. Our key insights, such as student involvement in plan granularity and semantic over syntax distillation, are rigorously validated and represent novel contributions to knowledge distillation. The extensive OOD evaluation confirm that DARGO distils transferable reasoning patterns rather than memorizing dataset-specific artifacts.

**Comparison with existing methods:**

We provide comprehensive comparisons with established distillation frameworks (DSPy, ReGAL) showing that DARGO consistently outperforms both approaches across all datasets In Table 1.

We thank all reviewers for their constructive feedback, which helped strengthen our work.

---

### Decision · Program_Chairs · 2025-09-17

**Decision:**

Accept (poster)

**Comment:**

### **The Key contribution of the paper:**
- **A Novel Framework for Knowledge Distillation**: The paper introduces DARGO (Distillation and Adaptive Reasoning-Guided Orchestration), a novel framework designed to transfer knowledge from a powerful Large Language Model (LLM) "teacher" to a more efficient Small Language Model (SLM) "student" without the need for intensive fine-tuning. The framework is particularly useful for Data Analysis Code Generation (DACG) tasks, where privacy is a concern.

- **Effective Use of In-Context Learning (ICL)**: It proposes a non-parametric knowledge distillation method that uses In-Context Learning (ICL), which is a training-free approach, to adapt SLMs for complex tasks. This approach is a significant alternative to traditional, resource-intensive fine-tuning, which often faces challenges due to limited high-quality training data and poor generalization across different programming languages or updates.

### **Summarization of Reviewer Comments**
The reviews for Submission 21305 are largely positive, with all three reviewers recommending acceptance. Reviewer xWYB initially provided a high score, while Reviewers Nr8J and nVTU were initially more reserved, citing concerns about experimental scope, engineering complexity, and presentation. The authors' detailed and data-driven rebuttal successfully addressed these weaknesses by providing clarifications, additional experimental results on recent models, and robust defenses of their design choices. The new data, which showed DARGO's consistent improvements on Qwen and other models, was particularly effective in swaying the reviewers. The overall consensus is that the paper presents a technically solid, valuable, and practical framework with strong empirical evidence, and the authors effectively handled the criticisms raised during the review period.

**For the response to Reviewer Y9rz**:
The authors provided a clear rebuttal to the reviewer's key concerns:
- **Cost**: The authors demonstrated that their DARGO framework is more cost-effective and delivers better performance compared to other distillation methods like DSPy and ReGAL.

- **Speed and Storage**: The reviewer raised concerns about potential bottlenecks from increased task volume. The authors clarified that the system's database is fixed during inference, ensuring that the per-query latency remains constant regardless of the total number of test queries. Although the reviewer dismissed this as "common sense," the authors' explanation confirms the system's speed advantage in its design.

- **Scope**: To justify their focus on Python/SQL, the authors highlighted the complexity and real-world significance of Data Analysis Code Generation (DACG), citing developer survey data to support the relevance and impact of their work.

In summary, the authors’ rebuttal effectively addresses the reviewer's technical criticisms by providing concrete evidence of their system's efficiency and a strong rationale for their research focus. While the reviewer remained unconvinced, the authors' explanations validate the paper's technical merits.